# Gender imbalances in the editorial activities of a selective journal run by academic editors

**Tal Seidel Malkinson**[1,2]*, **Devin B. Terhune**[3,4], **Mathew Kollamkulam**[5], **Maria J. Guerreiro**[6], **Dani S. Bassett**[7,8], **Tamar R. Makin**[5,9]

**1** Sorbonne Université, Institut du Cerveau - Paris Brain Institute - ICM, Inserm, CNRS, APHP, Hôpital de la Pitié Salpêtrière, Paris, France, **2** Université de Lorraine, CNRS, CRAN, F-54000 Nancy, France, **3** Department of Psychology, Goldsmiths, University of London, London, United Kingdom, **4** Department of Psychology, Institute of Psychiatry, Psychology and Neuroscience, King's College London, London, United Kingdom, **5** Institute of Cognitive Neuroscience, University College London, London, United Kingdom, **6** eLife Sciences Publishing Ltd., Cambridge, United Kingdom, **7** Departments of Bioengineering, Electrical & Systems Engineering, Physics & Astronomy, Neurology, and Psychiatry, University of Pennsylvania, Philadelphia, PA, United States of America, **8** Santa Fe Institute, Santa Fe, NM, United States of America, **9** MRC Cognition and Brain Sciences Unit, University of Cambridge, Cambridge, United Kingdom

* tal.seidel@mail.huji.ac.il

**Data Availability Statement:** All data are available from the Open Science Framework (OSF) repository (https://osf.io/c8a3t/).

## Abstract

The fairness of decisions made at various stages of the publication process is an important topic in meta-research. Here, based on an analysis of data on the gender of authors, editors and reviewers for 23,876 initial submissions and 7,192 full submissions to the journal eLife, we report on five stages of the publication process. We find that the board of reviewing editors (BRE) is men-dominant (69%) and that authors disproportionately suggest male editors when making an initial submission. We do not find evidence for gender bias when Senior Editors consult Reviewing Editors about initial submissions, but women Reviewing Editors are less engaged in discussions about these submissions than expected by their proportion. We find evidence of gender homophily when Senior Editors assign full submissions to Reviewing Editors (i.e., men are more likely to assign full submissions to other men (77% compared to the base assignment rate to men RE of 70%), and likewise for women (41% compared to women RE base assignment rate of 30%))). This tendency was stronger in more gender-balanced scientific disciplines. However, we do not find evidence for gender bias when authors appeal decisions made by editors to reject submissions. Together, our findings confirm that gender disparities exist along the editorial process and suggest that merely increasing the proportion of women might not be sufficient to eliminate this bias. Measures accounting for women's circumstances and needs (e.g., delaying discussions until all RE are engaged) and raising editorial awareness to women's needs may be essential to increasing gender equity and enhancing academic publication.

**Funding:** • Agence Nationale de la Recherche Award Number: ANR-16-CE37-0005 | Recipient: Tal Seidel Malkinson, Ph.D. • Wellcome Trust Award Number: 215575/Z/19/Z | Recipient: Tamar R. Makin • European Research Council Award Number: 715022 EmbodiedTech | Recipient: Tamar R. Makin The funders had no role in study design, data collection and analysis, decision to publish, or preparation of the manuscript.

## Introduction

Women remain underrepresented in science, technology, engineering, mathematics and medicine (STEMM), and are also prone to experiencing bias and discrimination [1–5]. This gender gap in representation and career advancement is present across all career stages [1, 6–9]. For example, beyond the clear disproportionate representation of men over women in senior investigator categories, women receive fewer and less prestigious awards [10–14], obtain fewer grants [15–17], are less frequently invited to write review or comment papers [18–21], and have lower salaries relative to men [6, 7, 22]. Gender disparities at senior levels are also noticeable for services to the broader scholarly community, where men are more likely to provide higher status external service, whereas women tend to perform lower status internal service [11, 23]. Moreover, although women and men spend comparable time at work, differences in how they fulfil their various responsibilities outside research (e.g., teaching and service compared with research) [24, 25] may contribute to differences in productivity and ultimately to other markers of career success [2, 8, 26, 27]. Due to these and other factors, women benefit from less prominence and eminence at senior levels, relative to men [2, 5, 11, 28]. These disparities can arise from structural, institutional, and systemic sexism as well as pervasive bias (whether implicit or explicit) harboured by colleagues of any gender [29–31], and can have multiple adverse implications (e.g., for women's pay [6, 7, 22] and promotion [1, 2, 6–8, 22]).

Scientific publishing is a central aspect of academia, with critical implications for hiring decisions and career advancement. Inequalities, based on an author's gender, have been systematically documented along different stages of the scientific publishing process [4, 20, 32–34]. First, the proportion of women as first and senior authors in peer-reviewed publications is lower than expected given their prevalence in the field [4, 20, 35–43]. Moreover, across different fields, women tend to submit fewer papers than men [43–45], with larger imbalances in journals with higher impact factors [46]. A higher publication standard for women authors, which in turn leads to decreased productivity, could contribute to this gap [47], as well as a smaller likelihood for attribution of credit in authorship for women than for men researchers [32, 33]. Gender inequities are also evident once women cross the submission hurdle, in the evaluation of women-led manuscripts [41, 47–49] [Though see 50 for opposite results]. For example, in several studies manipulating authors' identity, reviewers evaluated conference abstracts, papers, and fellowship applications supposedly written by men as better than when they were supposedly written by women [51, 52]. Moreover, a recent analysis of peer review outcomes of 23,876 initial submissions and 7,192 full submissions that were submitted to the journal eLife showed a homophily effect between reviewers and authors [53]. In particular, the acceptance rate for manuscripts with men senior authors was greater than for women senior authors and this disparity was greatest when the team of reviewers only comprised men [53]. After publication, women are less cited than expected [54–63, though see 64 for opposite results]. This imbalance is mainly due to a homophily effect in men authors, wherein men under-cite women's publications compared to men's publications [54, 65].

Gender disparities in the scientific publishing process may be further exacerbated by the underrepresentation of women among journal reviewers and editors. Editorial service is an essential element of the scientific enterprise. Editors and editorial boards are tasked with establishing benchmarks for scientific publishing and do so by engaging with a wide network of authors, reviewers, and other members of editorial boards. Insofar as editorial service has the potential to influence the progress and direction of a given scientific field, appointment to an editorial board reflects the high regard and trust of a community towards individual editors [5, 11, 28]. Despite repeated calls for making deliberate effort to incorporate gender diversity into editorial board structures [5, 66], gender disproportions remain pervasive [67–73]. Presently,

little is known about gender disparities in the editorial process itself. Here we address this knowledge gap by examining whether the involvement of individuals in an editorial board and along the different stages of the editorial process is subject to gender disparities.

We focused on the journal eLife, a non-profit open-access journal led by researchers, that aims to accelerate discovery by operating a platform for research communication that encourages and recognises the most responsible behaviours (https://elifesciences.org/about). eLife publishes selected research in all areas of biology and medicine, and its Editorial Board is structured to contain broad expertise required to evaluate research quality. eLife employs over 600 researchers in their Board of Reviewing Editors (BRE) and from 2019 onwards in particular have considered gender when recruiting new editors towards the goal of gender equality. eLife's review process broadly involves two main stages: initial evaluation of submissions by

the eLife editorial team, and evaluation of full submissions together with external reviewers (see Fig 1A). While the initial evaluation of submissions involves an internal consultation among eLife editors, the ensuing step of handling the review of full submissions includes community-facing interactions with external experts. eLife has been collecting meta data on all editorial interactions along this two-stage process, allowing to analyse not only women editors' representation in the editorial board, but also their active participation along the different stages, thus teasing apart potential versus actual engagement of women. For these reasons, eLife provides a rich case example to evaluate gender imbalance along key decision-making processes in STEMM and in particular in STEMM journals' editorial process.

The aim of this study was to determine whether the involvement of individuals in eLife's BRE is subject to gender disparities at various stages of the editorial process. Specifically, we sought to determine whether women eLife editors are proportionally involved in the editorial decision process compared to their representation in the BRE. To address this question, we explored fully anonymous analytics collected by eLife's editorial platform. This data was collected for monitoring purposes with the explicit aim to help improve eLife's submission and review process. The analytics provided binary gender information ("man" or "woman" as assigned by the editorial office based on scientists' names and perceived gender expression) relating to the handling of submissions. We assessed the presence of gender imbalance at different stages of editors' participation, starting with the external influence of authors who are invited to nominate potential editors (and appeal their decisions), through to the engagement of Reviewing Editors (REs) by Senior Editors, and then ending with the responsiveness of REs to editorial assignments. Based on the literature reviewed above [e.g., 53, 54, 67, 73], we predicted that despite efforts to increase the involvement of women in the BRE, women's editorial activities would be lower in comparison to men, even after taking into consideration their proportional disparity in the editorial system. Based on related research [53, 54, 67], we further predicted that decreased engagement would be exacerbated by a homophily effect, where men Senior Editors are more likely to engage men REs. By elucidating the editorial actions where gender imbalance is more prominent, we hope that this study will motivate the scientific community to work towards greater equity in this important process.

## Results

### Gender imbalance of eLife reviewing editors

We first quantified the gender ratio among eLife BRE members (Fig 1B). The proportion of RE service contributed by men was significantly larger than the proportion contributed by women throughout the entire study period (2017–2019: N = 12,518 months, women vs. men BRE service: 30.60% vs 69.40%; binomial $p < 0.001$; Cohen's $h$ = 0.40; see supplementary data and Sup Fig 1A for a comparison of months of service across gender).

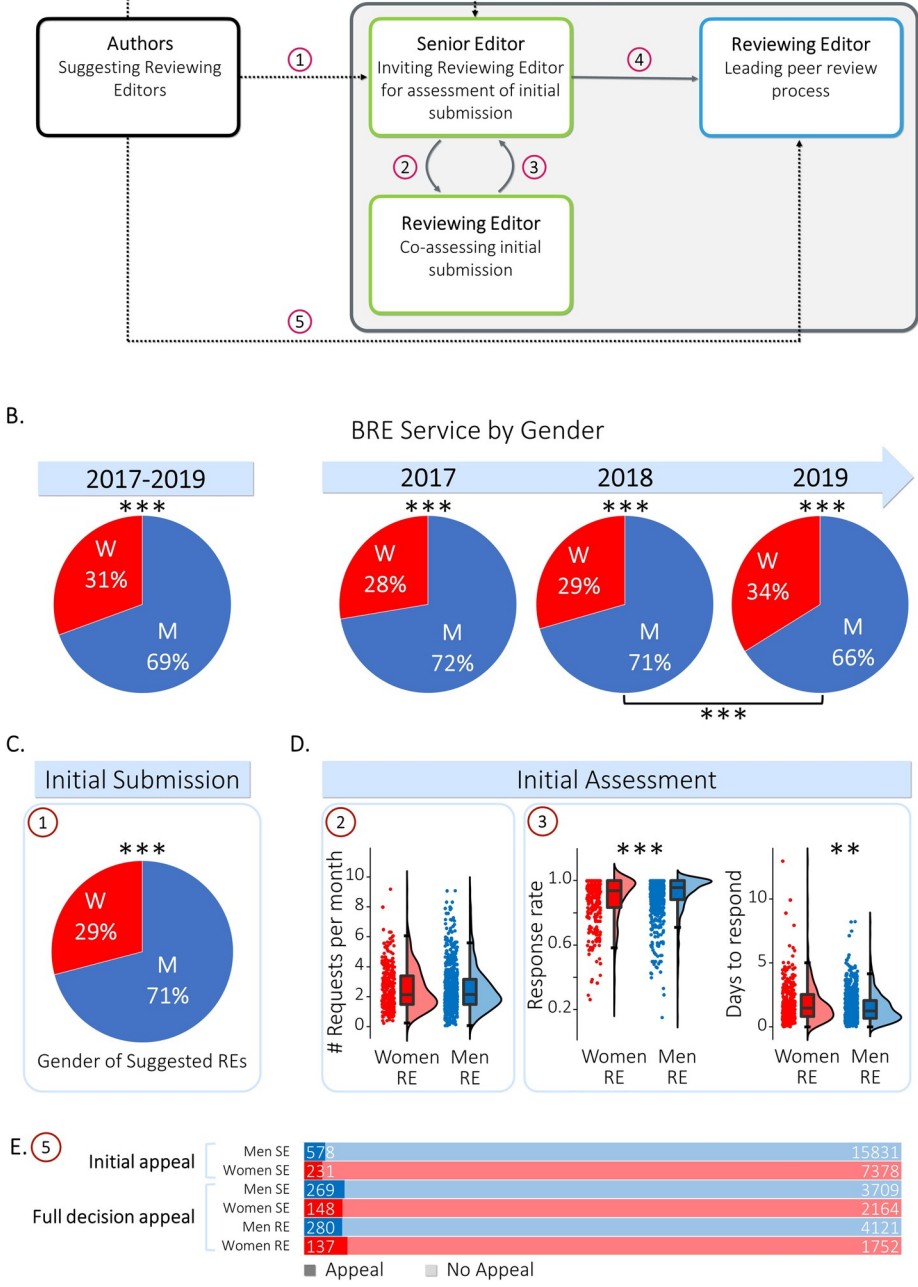

**Fig 1. Gender disparities in eLife's reviewing process. A.** A schematic of the locations along eLife's reviewing process wherein imbalanced actions could potentially occur (left to right): Initial Submission (Action 1)–Authors submit their manuscript and suggest potential members of the Board Reviewing Editors (BRE). Within eLife (grey square), a Senior Editor invites BRE members for initial consultation (Action 2) and the Reviewing Editor (RE) gives their opinion (Action 3). This stage of the editorial process is internal (green squares). Full Submission–If the manuscript is retained, the Senior Editor assigns a RE to lead the reviewing process (Action 4). This community-facing stage (blue square) includes overseeing reviewer selection and coordinating an open discussion between the reviewers, the handling Senior Editor and the RE once all individual reviewer reports have been submitted. Appeals–In the event of a rejection, Authors can appeal the initial assessment or the Full Submission decision (Action 5). **B.** Proportion of BRE service of women and men REs in the entire study period (2017–2019; left) and per study year (right). The gender disparity in BRE service is significantly imbalanced, as indicated by the asterisks. **C.** Gender imbalance in Initial Submission: Authors suggest more men REs than the men base rate when first submitting a manuscript (Action 1). **D.** Gender differences in Initial Assessment: Senior Editors equally engage women and men REs in the initial consultation

(Action 2). Women REs respond slightly less to Senior Editor's initial consultation requests (Action 3), and they take longer to respond than men REs. **E.** Appeal rates (Action 5) in the Initial Assessment (Senior Editors only) and Final Decision (Senior and Reviewing Editors) do not depend on the gender of the handling BRE. W = women (red); M = men (blue); SE = Senior Editor; RE = Reviewing Editor; Dashed arrows–Actions external to eLife, Full grey arrows–Actions within eLife; $*p \leq 0.05$, $**p \leq 0.01$, $***p \leq 0.001$.

We next considered dynamics in gender balance over the three-year window. The gender imbalance observed overall slightly diminished over time due to eLife's effort to recruit more women to the BRE. The proportion of women in the BRE did not significantly differ between 2017 and 2018 (1.81% difference, $\chi^2_{(1)} = 3.10$, $p = 0.078$, Cohen's $h = 0.86$). By contrast, the proportion in 2019 was significantly greater than that in 2018 (N-1 $\chi^2$ proportion comparison test; 2018 vs. 2019: 4.42% difference, $\chi^2_{(1)} = 19.67$, $p < 0.001$, Cohen's $h = 0.78$). Despite this slight improvement, the BRE gender base rate remained strongly imbalanced (2017: $N = 3,715$ months, women vs. men BRE service months: 27.64% vs. 72.36%; 2018: $N = 4,047$ months, 29.45% vs. 70.55%; 2019: $N = 4,756$ months, 33.87% vs. 66.13%; binomial $p$-values $< 0.001$; 2017: Cohen's $h = 0.46$, 2018: Cohen's $h = 0.42$, 2019: Cohen's $h = 0.33$). Accordingly, and for all subsequent analyses, the 2017–2019 data were pooled to increase statistical power. Taken together, these results indicate that there exists a pronounced gender imbalance in the BRE gender base rate.

In the next analyses, we used the gender ratio of women BRE members as the base rate to measure if women RE engagement was proportional to what was expected by their representation in the BRE.

## External influence in the initial submission–(Action 1)

At the Initial Submission stage, authors suggest potential BRE members that could handle their manuscript (Action 1). We tested if this action was (im)balanced according to gender by comparing the proportion of women REs that were suggested by authors relative to the women BRE member base rate. A $N$-1 $\chi2$ proportion comparison test revealed that authors suggest significantly fewer women REs than the corresponding proportion among eLife's BRE (29.08% vs. 30.6%, $\chi2(1) = 11.65$, $p < 0.001$, Cohen's $h = 0.90$; Fig 1C). We next sought to determine whether women's perceived expertise might be a partial explanation for authors' imbalanced RE suggestions. Specifically, previous research points at potential disparity with the broadness of term women and men use when communicating research [74]. Accordingly, we tested whether women and men REs differed in the number of keywords used to showcase their expertise. We found that women and men REs did not differ in their numbers of associated keywords (Women: 5.51±2.19; Men: 5.32±2.39; $t_{(581)} = 0.932$, $p = 0.352$). We next sought to determine whether a difference in the scope and reach of the keywords associated with women and men REs could contribute to authors' imbalanced RE suggestions. Accordingly, we quantified the number of PubMed search results for women and men BRE members' keywords. Specifically, we queried the number of PubMed publications associated with the string of keywords provided by each RE, using an 'OR' operator. This provided us with a simple mean of scope. A permutation Welch's $t$-test comparing groups in the number of PubMed search results was not significant (women vs. men search results: 1,755,724±2,979,049 vs. 1,920,643±3,307,501; $t(488.9475) = 0.62$; $p = 0.56$; Hedge's $g = -0.052$). These data provide no evidence of a gender difference in the overall reach of the keywords provided by BRE members.

## Internal processes in the initial assessment stage (Actions 2–3)

We next explored the presence of gender imbalances during Initial Assessments. In this action, the Senior Editor invites one or more REs for an initial consultation in order to assess whether to invite a full submission of the manuscript for peer review. To test whether Senior Editors tend to similarly engage women and men REs (Action 2), we compared the average number of consultation requests per month for individual REs. A permutation Welch's $t$-test showed no significant difference in the mean number of requests per month between women and men REs ($t_{(809.7)} = 0.11$, $p = 0.92$; Fig 1D), indicating no evidence for imbalanced engagement solely based on RE gender in this Action. While examining the distributions of requests per month, it appeared that the distribution of the men REs might have a longer tail (kurtosis men RE = 2.36; kurtosis women RE = 1.28). Intuitively, a gender difference in the distribution of requests per month could be due to the increased involvement of selected men REs. To examine this possibility, we selected the BREs who were disproportionally engaged in initial consultations relative to the BRE; that is, the 43 REs defined as the upper outliers of the population (defined as higher than the 75th percentile+1.5×interquartile range), with an average of 6.9 monthly consultations, relative to 2.24 on average. We find that only 10 of these especially engaged REs (23%) were women. However, Levene's test for equality of variances did not show significant differences between men and women RE request distributions ($F_{(1,1217)} = 0.052$, $p = 0.82$). As such, we find no evidence for gender differences when approaching REs for initial consultation.

We then evaluated the presence of gender differences in RE responses to the initial consultation request (Action 3). Compared to men REs, the response rate of women REs was significantly lower ($0.88\pm0.17$ compared to $0.91\pm0.14$; Welch's $t_{(651.4)} = 3.04$, $p = 0.001$, Hedges's $g = 0.20$; Fig 1D). In addition, women REs took longer to respond compared to men REs ($1.83\pm1.55$ days vs. $1.54\pm1.23$ days; Welch's $t_{(636.3)} = -3.24$, $p = 0.002$, Hedges's $g = -0.22$; Fig 1D). These data provide converging evidence for longer response time and less frequent responses of women REs when engaging in initial consultations, in comparison to men REs.

## Community-facing processes in the full submission stage (Action 4)

For manuscripts that pass the initial assessment, the Senior Editor assigns an RE who handles the reviewing process (Action 4). In order to evaluate the presence of imbalances in RE assignment, we first compared the number of full submissions per month handled by women and men REs. A permutation Welch's $t$-test showed that women REs handled slightly, though significantly, fewer submissions per month than men REs ($0.40\pm0.32$ vs. $0.44\pm0.37$; $t_{(869.8)} = 2.22$, $p = 0.026$, Hedges's $g = 0.13$; Fig 2A). We next explored the effect of the Senior Editor's gender on manuscript assignment to women and men REs. Using a contingency table analysis, we compared the proportion of manuscripts assigned to women and to men REs as a function of Senior Editor gender. Compared to RE gender base rates of manuscript assignment (6,289 manuscripts; women vs. men RE assigned manuscripts: 30.04% vs. 69.96 %), women Senior Editors assigned significantly more manuscripts to women REs (41.41% for women SEs vs. 30.04% for all SEs) and men Senior Editors assigned significantly more manuscripts to men REs (76.57% for men SEs vs. the 69.96% for all SEs; $\chi^2_{(1)} = 224.55$, $p<0.001$, contingency coefficient 0.186; Fig 2B). These results demonstrate that both women and men SEs are more likely to assign papers to REs of the same gender relative to the gender base rates.

In order to examine how this manifestation of gender homophily might vary across disciplines, we next divided the manuscripts according to the disciplines the authors assigned to their submission (up to 2 out of 18 suggested discipline categories; see Table 1). We repeated the contingency table analysis for each discipline separately and found a significant homophily

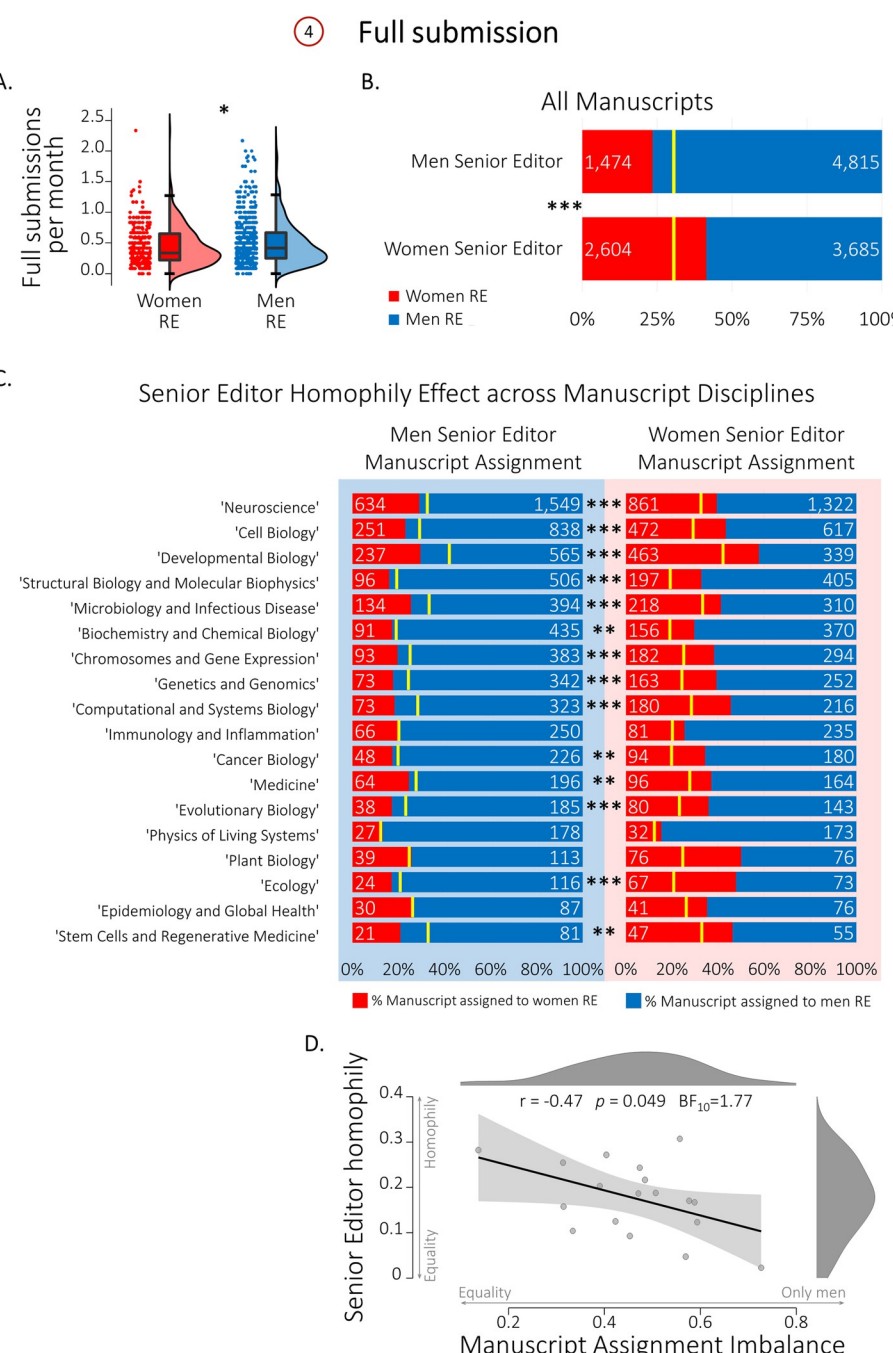

**Fig 2. Gender disparities in eLife during full submission (Action 4). A.** Men REs (blue) handle slightly more full submissions per month than women REs (red). **B.** Compared to manuscript assignment gender base rates (yellow lines; Base rate of RE manuscript assignment: Women– 30.04%; Men– 69.96 %;), Men Senior Editors (SE, top) assign significantly more manuscripts to men REs (blue; 76.57%) and women Senior Editors (bottom) assign significantly more manuscripts to women REs (red; 41.41%). **C.** SE-BRE Manuscript assignment homophily is prevalent across disciplines. The effect of Senior Editor gender on the assigned RE's gender across manuscript disciplines, showing preferential assignment of men REs (blue) by men Senior Editors (left) and of women REs (red) by women Senior Editors (right), compared to the gender base rate of RE manuscript assignment (yellow lines; $p$ values are FDR corrected). **D.** A scatter plot showing the correlation between the Senior Editor homophily effect (the difference in the rate of manuscripts assigned to men REs when the Senior Editor is a man and when the Senior Editor is a woman) and the Manuscript Assignment Imbalance (the difference in the rate of manuscripts assigned to men REs versus to women REs), across disciplines (Pearson $r = -0.47$, $p = 0.049$, BF10 = 1.77). Shaded area depicts the 95% confidence interval. $^*p \leq 0.05$, $^{**}p \leq 0.01$, $^{***}p \leq 0.001$.

**Table 1. Descriptive statistics and contingency table analysis of senior editor homophily effect across disciplines.** MS–Manuscripts; RE–Reviewing Editor; M–Men; W–Women.

| | Discipline | # MS | Base rate of M RE MS | M Senior Editor: % M RE MS | W Senior Editor: % M RE MS | $X^2_{(1)}$ | p *significant FDR corrected | Cramer's V |
|---|---|---|---|---|---|---|---|---|
| 1 | 'Neuroscience' | 2183 | 66.70 | 70.95 | 60.54 | 25.76 | <0.001* | 0.11 |
| 2 | 'Cell Biology' | 1089 | 69.51 | 76.96 | 56.64 | 49.23 | <0.001* | 0.21 |
| 3 | 'Developmental Biology' | 802 | 56.86 | 70.43 | 42.23 | 64.93 | <0.001* | 0.29 |
| 4 | 'Structural Biology & Molecular Biophysics' | 602 | 79.40 | 83.98 | 67.27 | 20.45 | <0.001* | 0.18 |
| 5 | 'Microbiology & Infectious Disease' | 528 | 65.72 | 74.57 | 58.78 | 14.39 | <0.001* | 0.17 |
| 6 | 'Biochemistry & Chemical Biology' | 526 | 79.66 | 82.66 | 70.31 | 9.12 | 0.003* | 0.13 |
| 7 | 'Chromosomes & Gene Expression' | 476 | 73.53 | 80.40 | 61.71 | 19.85 | <0.001* | 0.20 |
| 8 | 'Genetics & Genomics' | 415 | 74.22 | 82.31 | 60.65 | 23.81 | <0.001* | 0.24 |
| 9 | 'Computational & Systems Biology' | 396 | 70.20 | 81.66 | 54.49 | 34.08 | <0.001* | 0.29 |
| 10 | 'Immunology & Inflammation' | 316 | 78.48 | 79.25 | 74.51 | 0.57 | 0.451 | 0.03 |
| 11 | 'Cancer Biology' | 274 | 78.83 | 82.63 | 65.57 | 8.27 | 0.004* | 0.17 |
| 12 | 'Medicine' | 260 | 71.15 | 75.44 | 62.92 | 4.47 | 0.035* | 0.04 |
| 13 | 'Evolutionary Biology' | 223 | 75.34 | 82.84 | 64.05 | 10.16 | 0.001* | 0.21 |
| 14 | 'Physics of Living Systems' | 205 | 86.34 | 86.93 | 84.61 | 0.18 | 0.675 | 0.03 |
| 15 | 'Plant Biology' | 152 | 73.68 | 74.32 | 50.00 | 1.19 | 0.276 | 0.11 |
| 16 | 'Ecology' | 140 | 77.86 | 82.91 | 52.17 | 10.53 | 0.001* | 0.27 |
| 17 | 'Epidemiology & Global Health' | 117 | 72.65 | 74.00 | 64.71 | 0.63 | 0.427 | 0.07 |
| 18 | 'Stem Cells & Regenerative Medicine' | 102 | 65.69 | 79.17 | 53.70 | 7.31 | 0.007* | 0.27 |

effect of Senior Editor gender on the gender of the assigned RE in 14 out of the 18 disciplines (78%; Contingency table analysis with FDR correction for multiple comparisons; see Table 1 for details). Fig 2C shows the gender homophily in RE manuscript assignment across all manuscript discipline categories. These results demonstrate that gender homophily in manuscript assignment is a widespread cross-disciplinary effect.

Previous research suggests that homophily effects negatively associate with the extent of gender imbalance [54]. Accordingly, we next explored associations between homophily in manuscript assignments and gender across disciplines (Fig 2D). For each discipline, we first defined Senior Editor homophily as the difference between (i) the proportion of manuscripts assigned to men REs by men Senior Editors and (ii) the proportion assigned to men REs by women Senior Editors. Intuitively, a value of zero indicates no gender difference between men and women Senior Editor manuscript assignments, whereas a value of unity indicates that Senior Editors only assign manuscripts to REs of their own gender. We similarly defined for each discipline an index that we refer to as the *manuscript assignment imbalance*, which is calculated as the difference between (i) the proportion of manuscripts assigned to men REs and (ii) the proportion assigned to women REs. Intuitively, a value of zero indicates a fully balanced discipline, whereas a value of unity indicates that manuscripts are assigned exclusively to men REs. Across disciplines, the correlation between Senior Editor homophily and the manuscript assignment imbalance index was negative, albeit borderline in statistical significance ($r = -0.47$, $p = 0.049$). A Bayesian correlation analysis also suggested only anecdotal evidence in favour of a negative association ($BF_{10} = 1.77$). This result provides preliminary evidence that in disciplines with more equal manuscript assignment, Senior Editor homophily is stronger, in line with previous research [54]. http://www.python.org).

### Appeals (Action 5)

In our final analysis, we evaluated the presence of gender imbalances in authors' appeals (Action 5). In the Initial Assessment stage, only the identity of the Senior Editor is revealed to the authors. The difference in the rates of appeals of manuscripts handled by women and men Senior Editors in the Initial Assessment was marginal; we observed a trend towards fewer appeals over women Senior Editors' assessments, but this trend did not reach statistical significance (Contingency table analysis, $\chi^2_{(1)}$ = 3.781, $p$ = 0.052, Contingency coefficient = 0.013; Fig 1E). Moreover, a Bayesian Contingency table analysis suggested moderate evidence in favour of the null hypothesis ($BF_{10}$ = 0.28), confirming the lack of difference in Senior Editor gender in Initial Assessment appeals. In the Full Submission stage, the identities of both the Senior Editor and the handling RE are revealed to the authors. Dovetailing with the Initial appeals findings, the gender difference in the rates of appeals for the final decision for both Senior Editors and REs did not reach significance (Senior Editor gender: Contingency analysis, $\chi^2_{(1)}$ = 0.34, $p$ = 0.58; RE gender: Contingency analysis, $\chi^2_{(1)}$ = 1.69, $p$ = 0.19; Fig 1E). These results suggest that in general, authors' tendency to appeal does not seem to depend on the gender of the Senior Editor and handling RE. It is important to note, however, that the small rate of appeals limits the robustness of this finding: we observed 809 initial assessment appeals out of 24018 initial submissions (3.4%), and 417 final decision appeals out of 6289 fully submitted manuscripts (6.6%).

## Discussion

Gender imbalance in the scientific publishing process is already evident when considering simple numerical disparities, starting with women's representation in scientific editorial boards [67–73], number of invited articles [18, 19, 21], frequency of being asked to referee [75–77], published manuscripts' topics [34], and number of publications [36, 37, 78]. Here we extend the scope of this disparity by reporting clear under-representation of women in the BRE of a prominent biomedical journal (eLife). Beyond numerical proportions, the eLife dataset allowed us to examine whether the various actions that make up the editorial process are related to RE gender. We find that gender disparity stretches well beyond the known numerical imbalance, hinting at gender biases influencing the editorial process. Moreover, in a number of cases, gender disparity effects were large in magnitude. The gender disparity is first exerted by external influence—authors suggest more men from the pool of REs, even after correcting for men's numerical over-representation in the BRE. We also see gender disparity within eLife, in terms of the RE's bidirectional engagement during the internal initial assessment of submissions. Perhaps most strikingly, we find a robust homophily effect when assigning REs to lead the community-facing role of the editorial peer review. Each of these gender disparity effects is compatible with previous research demonstrating systematic biases in STEMM. Where we add to this body of knowledge is by uncovering the internal working of editorial decisions that will impact the participation and contribution of women. By revealing multiple contributing factors that exacerbate the existing imbalance, our findings highlight the need to assess and correct gender disparities in terms of the *contribution* to the editorial process (equity) and not just in terms of proportional *representation* (equality). It is our hope that a better understanding of these mechanisms will help reduce the biases that we document.

### The eLife dataset

Before we discuss our key findings, it is important to consider our unique dataset and the potential advantages and limitations inherent to it. As detailed in the Methods section, we used anonymous analytics collected by eLife's editorial platform for monitoring purposes.

This rich dataset reflects a real-life process, and spans a relatively large range of biological disciplines and international contributing scientists and editors. During the investigation period, eLife had a similar fraction of women in their BRE relative to other editorial boards [73], suggesting that the issues identified here are likely to be observed in other journals. However, the specific factors that we could study were not pre-determined based on our experimental needs. Accordingly, we were limited in our explanatory power, both in terms of other relevant factors that might be contributing to the observed effects (e.g., the level of seniority of each RE) and in terms of the statistical power (e.g., authors' appeals are rather infrequent). To mitigate some of these gaps, we can gain some insight from more recent data relating to the heterogeneity of eLife's BRE (see Supplemental Section, S1 Fig), although these recent analytics may not fully represent the dataset we analysed here. It is also important to consider the makeup of the BRE; these are invited roles, and as such, all the REs are established in their subfields. However, due to issues we expand upon below, it is possible that women REs are less senior than men REs, as described in the Supplemental data. Unfortunately, the fact that our primary datasets lack direct information on academic attainment levels for all women and men is a limitation. This lack of information should not be interpreted to mean that academic status is equal across genders in our datasets, an assumption that is likely to be incorrect. The sociology of gendered behavior predicts that both academic status and gender likely influenced the outcome of the interaction in which manuscripts were evaluated, as discussed below. We also do not have data on the intersectionality of gender with other primary sources of disparity (e.g., geographic location, race, ethnicity, class, sexual orientation, and ability [79–82]). Yet, the results of a recent eLife self-report survey conducted outside our study period suggest that women serving as editors are more likely to also self-identify as belonging to an underrepresented or minority group based on their race or ethnicity [83]. Finally, as described in the Methods section, perceived gender was assigned as "man" or "woman" (without distinguishing trans from cis) based on the REs' names and public profiles, and hence may or may not reflect the BRE's true gender identity. Although eLife recent data suggest that the vast majority of the BRE is cis [83], gender identity was not measured along and outside the binary (e.g., nonbinary, genderfluid, etc.). With these points in mind, our gender effects might be modulated by other contributing factors that should be investigated in future research in greater detail.

## Gender disparities

We first considered gender differences in REs bidirectional engagement, including both invitations to contribute to the initial editorial consultation by the Senior Editors and the individuals' participation in response. We define this process as internally-facing because the identity of the REs involved is only revealed to the other editors engaged in the consultation. We did not find significant differences in the number of invitations of women REs by the Senior Editors to participate in initial consultations relative to men REs. However, we did observe a heavy-tailed, skewed distribution of consultations, such that there is a small group, mostly comprised of men, that disproportionally dominates initial consultations. Even if the differential proportions of these groups are not statistically significant, this small men-dominated group might still skew diversity [84, 85]. To distribute the influence more fairly, a potential solution is to cap the number of consultations per individual RE.

Although the number of initial invitations did not differ between women and men, women engaged less with invitations from the SEs, resulting in the under-involvement of women in editorial activities. Women took only slightly longer to respond relative to men (women were approximately 7 hours slower to answer emailed invitations), but considering the interactive nature of the consultation process, this delay could be meaningful. In the eLife initial

consultation process, where editors' interact in an on-line instant chatting format, this means that men are more likely to set the tone of the discussion by providing their opinion first, making it more difficult for women, on average, to influence the editorial decision (through conformity and anchoring cognitive biases for example [86–88]). It has been previously shown that it is more difficult to voice a different opinion once an opinion has been formed [89, 90]. The delayed response, as well as reduced response rate (by approximately 3%) could potentially be attributed to the fact that women have more duties and responsibilities than men REs. There are multiple reasons to suggest this, depending on women's specific intersecting identities [43, 91, 92]. For example, senior women are overburdened by administrative responsibilities due to the institutional need to narrow the gender gap [25, 30]. More specifically to our dataset, there is a hint that women REs are at an earlier career stage relative to men (Supplementary section), and hence may be more likely to have children at home than their men colleagues and thus face an added burden on their time [43], or be more laden with obtaining tenure. Another potential contributing factor is the higher standard of communication women are held to in order to receive equal acknowledgment, resulting in an imposed time-consuming quantity/quality trade-off for women, and reducing their productivity [47, 93, 94]. Irrespective of the reasons, our results signal that the journal submission and review process needs to shift away from monitoring decisions based on the decision time, which adds time pressure, and instead could potentially delay discussion and/or decisions about submissions until women have contributed.

We did find a significant difference in the engagement of women REs when considering community-facing duties, particularly when leading the peer-review process. Specifically, women were assigned 9% fewer manuscripts relative to men. This effect is likely exacerbated by the longer response time and less frequent responses we observed during initial consultations, as the assignment of the reviewing RE is often determined during the initial consultation. We are hopeful that if the bias in the previous stage is corrected then the under-assignment of full submissions to women REs will be improved. However, it is also important to consider more carefully other potential sources of bias and how to mitigate them. For example, it is also possible that men might volunteer more readily to take up this time-consuming role–our data does not allow us to shed any light on the inner discussions beyond response time. Regardless, our effect is consistent with other studies showing that women are disproportionately engaged in internal-institutional facing duties, whereas men are disproportionately engaged in community-facing roles, which are also more associated with eminence, networking, and other benefits related to the more visible duties of the reviewing RE leading the peer review [11, 12, 14]. The reasons underlying this pattern should be further studied, however women's different time allocation may reflect a purposeful choice to contribute to their institutions. Another potential driver could be inherent biases of the Senior Editor assigning the RE; research shows that women are less frequently approached to apply for awards, write invited reviews, etc. [11, 13, 19, 21, 75]. Within the context of editorial assignments, this effect could be potentially corrected by providing gender-specific statistics to the Senior Editors about disproportional engagement by gender. We turn to consider gender-based interactions between the Senior Editor and REs in the next section.

## Homophilic behaviours

Homophily is one of the fundamental patterns underlying human relationships across multiple social systems, influencing how communities form, how status is distributed, and how sub-groups evolve in occupations and organizations [95]. With respect to the homophily effect of the Senior Editor's gender on REs assignments, we find that across multiple sub-disciplines,

there is a significant tendency for Senior Editors to choose same-gender REs to handle full submission for peer reviews. One might wonder whether the observed homophily effects might be explained by field-specific differences in gender proportions: in a discipline comprised mostly by men, e.g. physics of living systems, the Senior Editor (likely a man) will more often reach out to more men simply because most of the experts are men. To evaluate this possible explanation, we separated our data by discipline. We found that the homophily effect exists quite broadly, across 14 of the 18 disciplines (despite noticeable variability in the proportion of women/men RE across disciplines, see Fig 2C), hence refuting the differently-gendered subdisciplines account. What other drivers could potentially explain the homophily effect? Homophily is driven by various types of associations and dimensions of similarity [96], such as ascribed attributes (e.g. gender [97]), acquired attributes (e.g. occupation [98]), values, attitudes, and beliefs (e.g. activism [99]). Homophily, and gender homophily in particular, are prevalent in academia, for example in shaping interactions in scientific conferences [100], affecting scientific collaboration and scientific societies [11, 101], and biasing the selection of Nobel laureates [13]. Thus, we were not surprised to find that men Senior Editors assign more men REs than the women REs, even after taking into consideration the larger numerical proportion of men in the BRE.

It is possible that homophily in women arises from different drivers than homophily in men [67, 102], due to distinct social processes [103, 104] and the roles they play in intersectional power structures [105]. Considering the current political climate where there is greater awareness for the under-representation of women in STEMM, it is possible that women Senior Editors adopt an informal policy to engage women REs disproportionately. In this respect, the women homophily offsets to some degree the gender bias we see in the editorial process. Activism-driven homophily among women was demonstrated for example in crowdfunding of start-up projects, whereby a small proportion of women backers disproportionately supported women-led projects in areas where women are historically underrepresented [99]. Similarly, gender homophily in reviewer assignment by journal editors was widespread among men editors, while for women only a small number of highly homophilic editors dominated [67]. Our data did not allow us to directly explore the prevalence of homophily among individual REs, yet the fact that homophily was widespread across many fields, involving different REs, suggests women homophily is a broad phenomenon in eLife. Additionally, we find that homophily increases with gender balance across sub-disciplines. This echoes the finding that men homophily in article citations increases as the research field gets more gender balanced with time [54]. However, given that women Senior Editors are outnumbered by men (for example, 36% (30) women vs. 64% (52) men Senior Editors in 2021), on average we see an over-engagement of the men REs, even after accounting for their numerical dominance in the BRE. One simple candidate intervention is to increase the proportion of women in senior roles, which could also potentially serve to address other aspects of gender disparity that we did not study here. However, for the reasons detailed above, simply increasing representation (e.g., the number of women) might not be sufficient to ensure inclusion, equity, and justice [11, 67, 99, 106, 107].

Despite the fact that women display homophilic tendencies that serve to partly balance the homophilic tendencies of men, we do not in general endorse homophily effects as an appropriate solution to the gender bias observed here, as it can have devastating trickle-down consequences. For example, it was previously shown that scientific journal editors of both genders were more likely to appoint reviewers of the same gender as themselves [67]. Moreover, a previous study of eLife editorial decisions focused on how the gender makeup of the participants in the peer-review stage–both editors and reviewers–biases acceptance rates for men and women authors [53]. It was observed that all-men reviewer teams are far more likely to accept men-led manuscripts. Therefore, the homophilic behaviour that we observe among men is

likely to exacerbate these effects and increase the gender publishing gap. More generally, it was shown that homophilic groups tend to have similar evaluations and mind-sets [67, 108, 109]. Hence, the uncontrolled effects of homophily may undermine the impartiality of peer-review, and thus undermine science [67, 110]. Instead, solutions should be driven by formal policy that foreground equity and justice. For example, the homophily factor could be monitored to help Senior Editors avoid implicit and explicit biases. Another important candidate intervention for this issue is to diversify the network of the Senior Editors within the BRE.

## Conclusion

Table 2 provides a summary of our results and aims to offer potential guidance to stakeholders for taking a proactive approach towards enhancing gender equity in editorial activities. We find multiple consistent disparities across the editorial process, which culminate in the disproportional handling of submissions for peer review by men relative to women, even after taking into consideration men's over-representation in eLife's BRE. This effect was not a mere consequence of different gender distribution across disciplines, meaning it is not due to lack of available expert women, but rather a tendency of men SEs to favour men REs over women REs. This homophily effect is known to influence editorial decision-making, e.g. in recruitment of reviewers [67] and in favourable evaluation of manuscript led by men [53]. Therefore, it is easy to speculate that the disparity effects we observed here would be further amplified as the decision process trickles down. In other words, the gender disparity that tends to disfavour women cannot be pin-pointed to a single stage in the editorial evaluation process, but should instead be viewed as a systematic accumulation of biases across multiple decision-making steps of a people-led process.

To conclude, at the time of our analysis, eLife and other scientific journals do not have a formal strategy for engaging women, beyond increasing their numerical proportion. By including more women in the editorial process, the hope is that their voice will be expressed and heard. However, the evidence provided here suggests that simply increasing women's numbers is not enough to overcome gender bias. Critically, without taking into consideration women's specific work habits and availability, starting with their potentially different career demands, through different work-life balance and ending with sociological preferences, it is difficult to imagine a future in which the underlying mechanisms for under-engagement of women do not continue to bias the process. We therefore suggest that in order to index gender balance, we need a focus on equity rather than equality. We further suggest that informal policies, such as gender homophily, need to be replaced by formal policies that are based on educating both Senior and Reviewer Editors on how the choices that they make during editorial activities impact the gender gap.

## Materials and methods

In this methods section, we first provide a detailed description of eLife's peer review process, before describing the data we study and the statistical methods we employ.

### eLife's peer review process

eLife holds a unique two-stage evaluation process, as detailed in Fig 1A. The first stage is the initial assessment, and the second stage is peer review. We will describe each in turn, along with the series of actions it comprises.

**Initial assessment stage.**   In the first stage, submitted manuscripts are evaluated by a team of editors with related expertise. A Senior Editor solicits the advice of one or several REs in order to determine whether the manuscript is suitable for peer review. The process of soliciting

**Table 2. Summary of the study's main findings, speculated causes, and potential solutions.** Notice that the effects reported here were observed even after taking into consideration the reduced numerical representation of women in eLife's editorial system. These proposed solutions aim to provide potential guidance to stakeholders, enabling them to adopt a proactive and practical approach towards enhancing gender equity in editorial activities.

| Effect | Potential drivers | Recommendation |
|---|---|---|
| Authors suggest more men REs | Explicit or implicit bias /cultural norms/ internalised stereotypes/ differences in visibility | eLife can request authors to suggest a balanced gender representation and alert authors for disproportionate recommendation |
| Initial consultations disproportionately involve a subgroup of REs, mainly men | Explicit or implicit bias /cultural norms/ internalised stereotypes/ differences in visibility | Cap the number of consultations per individual RE to distribute influence more fairly |
| Women REs take slightly longer to respond to initial consultations; Women REs respond slightly less frequently to initial consultations | Women are held to a higher standard of communication/more affected by other commitments | Decision time should not be a limiting factor, reveal feedback after all REs had an opportunity to engage; Include more women in initial consultation to account for their lower response rate |
| Women handle fewer full submission | Explicit or implicit bias /cultural norms/ internalised stereotypes/ differences in visibility | Offset bias in initial consultation, provide feedback on gender imbalance patterns for Senior Editors (e.g., statistics about disproportionate RE engagement by gender), diversify the network of the Senior Editors within the BRE |
| Homophily effect | Same-gender network; Attempt to correct societal confounds | Increase transparency and awareness to the risks of homophily in science, increase the proportion of women Senior Editors, diversify the network of the Senior Editors within the BRE |

and receiving advice is carried out in an interactive consultation forum between all involved participants. Thus, the role of the RE at this stage is internal. The outcome of this process is communicated to the author in a letter signed by the Senior Editor. As such, the identity of the advising RE(s) is only known internally. To help the Senior Editor identify the most relevant members of the BRE to solicit as an advising RE, the authors are invited to suggest REs as part of their initial submission.

**Peer review stage.** For papers that are invited for full review, an RE is chosen to manage the process by overseeing the reviewer selection and by coordinating an open discussion between the reviewers, the handling Senior Editor, and the RE once all individual reviewer reports have been submitted. The RE is also encouraged to provide their own independent review as one of the peer reviewers. The RE facilitates the discussion and drafts a final decision either rejecting the paper or requesting the necessary revisions to support the acceptance of the paper. The identity of the RE is revealed not only to the reviewers in the discussion, but also to any other experts that were invited to take part in the peer-review process. Both Senior and Reviewing Editors sign the decision letter, and if the paper is published with eLife, they are also named as editors on the published manuscript. As such, the role of the RE at this stage is community-facing.

**Post-rejection.** In the event that a paper is rejected at either stage of the editorial process, the author(s) can appeal the editorial decision.

## Data

Data accumulated by eLife's platform for science publishing over the years 2017–2019 were organised into two datasets, as summarised in Table 3. The first dataset will be referred to as the BRE dataset, and the second will be referred to as the Manuscript dataset. We will describe each in turn. But first we make a note on assigned gender.

*Gender assignment.* In all cases, Editor gender was assigned by eLife's staff based on the editor's name and gender expression. Note that staff (i) assigned a binary "man" or "woman" gender, (ii) did not distinguish between trans and cis identities, and (iii) did not assign other genders such as nonbinary, genderqueer, agender, or genderfluid. Note that any editor could

**Table 3. eLife datasets.** Top: BRE Dataset: contains information relating to the engagement of individual BRE members in the editorial process (identified by gender and year). It includes the following fields: The mean number of days until the Reviewing Editor (RE) responded to a Senior Editor's request to participate in the Initial Assessment stage (Days to respond); The RE response rate to Initial Assessment consultation requests (Response Rate); The mean number of consultation requests per month each RE received (# Requests per month); The mean number of full submissions per month each RE handled (# Full submissions per month); The keywords associated with each RE to showcase their expertise (Keywords). Note that the number of full submissions may contain papers that the REs had handled as Guest Editors in the year prior to joining the BRE. Also, some REs may have been on leave, and therefore may have not been consulted for a certain period. Bottom: Manuscript Dataset: contains information relating to each manuscript submission, detailing the manuscript's outcome in each of the reviewing process stages (identified by gender of the Senior and Reviewing Editors). It includes the following fields: The proportion of men BRE members suggested by the authors (% of Men BRE members); The gender of the Senior Editor handling the manuscript throughout the reviewing process (Gender of Senior Editor); The gender of the RE handling the manuscript in the Full Submission stage (Gender of handling RE); The rate of author appeals at the Initial Assessment stage in which only the Senior Editor identity is revealed to the authors (Initial appeal rate); The rate of author appeals at the Full Submission stage in which both the Senior and Reviewing Editors' identities are revealed to the authors (Initial appeal rate); The two discipline terms the authors chose, out of 18 possible terms (Discipline 1 & Discipline 2; see Table 1 for details).

| BRE dataset | | | |
|---|---|---|---|
| **Year** | **2017** | **2018** | **2019** |
| **N** | **328** | **376** | **497** |
| **Days to respond** | | | |
| Mean | 1.517 | 1.613 | 1.723 |
| Std. Deviation | 1.165 | 1.277 | 1.49 |
| Minimum | 0.022 | 0.014 | 0.025 |
| Maximum | 7.958 | 8.902 | 12.974 |
| **Response rate** | | | |
| Mean | 0.915 | 0.897 | 0.899 |
| Std. Deviation | 0.114 | 0.15 | 0.162 |
| Minimum | 0.389 | 0 | 0 |
| Maximum | 1 | 1 | 1 |
| **# Requests per month** | | | |
| Mean | 2.462 | 2.336 | 2.419 |
| Std. Deviation | 1.471 | 1.421 | 1.549 |
| Minimum | 0 | 0 | 0 |
| Maximum | 9.083 | 9.091 | 9.2 |
| **# Full submissions per month** | | | |
| Mean | 0.463 | 0.415 | 0.41 |
| Std. Deviation | 0.374 | 0.343 | 0.349 |
| Minimum | 0 | 0 | 0 |
| Maximum | 2.333 | 2 | 1.9 |
| **Keywords** | | | |
| Manuscript dataset | | | |
| **Year** | **2017** | **2018** | **2019** |
| **N** (Total = 24056) | 7514 | 7670 | 8872 |
| **Full submission** (Total = 6289) | 1976 | 1948 | 1413 |
| **% of Men Suggested BRE members** | | | |
| **Gender of Senior Editor** | | | |
| **Gender of handling RE** | | | |
| **Initial appeal rate** (only Senior Editors) | | | |
| **Full decision appeal rate** (Senior Editors and REs) | | | |
| **Discipline 1** (18 possible disciplines) | | | |
| **Discipline 2** (18 possible disciplines) | | | |

have a gender different from the one that was assigned, and that true gender may or may not be more widely known by the community for several reasons: (i) scientists might be closeted due to the pervasive violence and discrimination faced by gender minorities, (ii) scientists might share their true gender identity only with a few close colleagues or friends, or (iii) scientists might share their identity freely but because of the complexity of the social network landscape in science, that information may not have reached all other scientists in their field. Accordingly, the staff's assignment of gender therefore reflects not self-identity but rather the perceived binary gender of the person. This perception is likely to also be held by the majority of the broader community, and hence is particularly relevant to understanding how the editor might be treated by that community (e.g., the frequency with which they might be suggested as a Reviewing Editor by authors). We also note that since early 2020, eLife has given all Senior and Reviewing Editors the option of sharing their self-reported gender identity via a confidential survey. However, the current response rate (~40%) precludes a comprehensive analysis of gender disparities using the data at this stage [83].

*BRE dataset*. This dataset includes anonymous information relating to the engagement of individual REs in the editorial process. This information includes the start and end dates of their editorial contracts, the number of consultations in which they have been invited to participate, how responsive they are to consultation requests (number of responses and response rate), the number of full submissions assigned, and how many days they take to make an editorial decision. In addition, the editorial staff asks REs to provide a set of keywords that reflect the scope of their research, which was also included in this dataset. Note that the terminology was self-generated by the REs (rather than adopted from an existing database), and that there were no limitations on the number of keywords each RE could provide. For some REs, additional keywords are added by the editorial staff based on the information publicly available on the editors' academic websites.

*Manuscript dataset*. This dataset includes information relating to each manuscript submission, detailing the manuscript's outcome in each of the submission stages. This dataset also contains the assigned gender (as described above) of those BRE members that were suggested by the authors, the recorded gender of the handling RE, and the recorded gender of the assigned Senior Editor. Note that here our information regarding gender pertains only to the editorial team handling the manuscripts and not to the manuscript authors, whose identities were not made available for the present study due to ethical considerations (though we note that the authors' identity, but not necessarily their self-defined gender, was known to the editors involved in the assessment). Manuscripts with appeals received after the Initial Submission and without a Full Submission decision were most likely rejected prior to review. It is possible that a small fraction of manuscripts was withdrawn prior to evaluation; however, we did not have access to such data.

Additionally, this dataset contains up to two (out of 18) disciplines that the authors assigned to their manuscript upon submission. Options included 'Neuroscience', 'Cell Biology', 'Developmental Biology', 'Structural Biology and Molecular Biophysics', 'Microbiology and Infectious Disease', 'Biochemistry and Chemical Biology', 'Chromosomes and Gene Expression', 'Genetics and Genomics', 'Computational and Systems Biology', 'Immunology and Inflammation', 'Cancer Biology', 'Medicine', 'Evolutionary Biology', 'Physics of Living Systems', 'Plant Biology', 'Ecology', 'Epidemiology and Global Health', and 'Stem Cells and Regenerative Medicine' (Manuscript Dataset; see Tables 1 and 2). In order to analyse the manuscript data across disciplines, we assigned to each discipline all the manuscripts in which a discipline was chosen at submission. This process created some overlap between disciplines (6289 fully submitted manuscripts; 1979 manuscripts were assigned to two disciplines out of 8268 assigned manuscripts, or 23.9%).

## Data analysis

We applied several exclusion criteria to the data before proceeding with further analysis. In the BRE dataset, we excluded REs who became Senior Editors, or resigned as Senior Editors (and became REs) in a given year, or those who were inactive (i.e., were never contacted on initial submissions). In addition, in the manuscript dataset, we limited the number of author-suggested REs to five per manuscript; and excluded papers handled by guest editors as well as Research Advances, Registered Reports, and formats that go through a different workflow.

## Statistical analysis

Results are reported as mean ± standard deviation (StD). Owing to several non-normal distributions in the data, we used non-parametric tests in all analyses. Binomial tests and $N$-1 $\chi^2$ proportion comparison tests were performed to compare one or two proportions using JASP (JASP Team (2020) Version 0.14) and MedCalc online tools (MedCalc Software, Ostend, Belgium), respectively. Contingency table analysis was used for testing the interrelation between binary variables using JASP software. When comparing the means of two groups with unequal sample sizes, we used a permutation-based Welch's independent $t$-test (10,000 permutations) in MATLAB (PERMUTOOLS package, The Math Works, Inc. MATLAB. Version 2020a, The Math Works, Inc., 2020. Computer Software. www.mathworks.com/). Pearson correlation coefficients were computed using JASP in order to test the association between continuous scale variables, after checking for normality assumption violations using the Shapiro-Wilk test for bivariate normality. When relevant, all tests were conducted using 2-tailed tests. We used Hedges's $g$ ($g = \frac{\overline{X_1} - \overline{X_2}}{SD^*_{pooled}}$) to compute effect size when comparing two means in a permutation Welch's t-test, and Cohen's $h$ ($h = 2\sin^{-1}\sqrt{p_1} - 2\sin^{-1}\sqrt{p_2}$) when comparing two proportions in a N-1 $\chi$2 proportion comparison test. When effects were close to the critical alpha ($p<0.05$), we conducted equivalent Bayesian analyses, with default prior settings (Bayesian correlation stretched beta prior width = 1; Bayesian Contingency tables, prior concentration = 1) using JASP to test whether there was more evidence for $H_0$ or for $H_1$. In order to measure BRE members service contribution (BRE service) as a function of gender, we computed the number of months in which the RE was affiliated with the BRE per year, i.e. the proportion of months of service of women and men REs out of the total number of service months, thus accounting for variability in BRE service contract durations and partial work time (e.g. REs appointed in the middle of the year, being on leave).

To investigate whether gender disparities were associated with REs' expertise, as advertised by eLife to prospective authors, we conducted an analysis of the relative scope and reach of the REs' keywords, broken down by the recorded gender of the RE. Keywords for each RE were extracted and strung together using the 'OR' operator and then queried against the PubMed database through NCBI's public API—'Entrez Programming Utilities (E-utilities)' (Entrez Programming Utilities Help [Internet]. Bethesda (MD): National Center for Biotechnology Information (US); 2010-. Available from: https://www.ncbi.nlm.nih.gov/books/NBK25501/). The number of search results for each set of RE keywords was recorded and used as a measure of the reach of the keywords provided by the REs, as evidenced by published papers related to the keywords in the literature. The E-Utilities API was accessed through a script in Python (Python Software Foundation. Python Language Reference, version 3.9.6. Available at

## Diversity statement

Recent work in several fields of science has identified a bias in citation practices such that papers from women and other minority scholars are under-cited relative to the number of

such papers in the field [54, 56–63]. Here we sought to proactively consider choosing references that reflect the diversity of the field in thought, form of contribution, gender, race, ethnicity, and other factors. First, we obtained the predicted gender of the first and last author of each reference by using databases that store the probability of a first name being carried by a woman [54, 111]. By this measure (and excluding self-citations to the first and last authors of our current paper), our references contain 30.62% woman(first)/woman(last), 22.82% man/woman, 18.14% woman/man, and 28.42% man/man. This method is limited in that a) names, pronouns, and social media profiles used to construct the databases may not, in every case, be indicative of gender identity and b) it cannot account for intersex, non-binary, or transgender people. Second, we obtained predicted racial/ethnic category of the first and last author of each reference by databases that store the probability of a first and last name being carried by an author of color [112, 113]. By this measure (and excluding self-citations), our references contain 6.21% author of color (first)/author of color(last), 15.01% white author/author of color, 16.03% author of color/white author, and 62.75% white author/white author. This method is limited in that a) names and Florida Voter Data to make the predictions may not be indicative of racial/ethnic identity, and b) it cannot account for Indigenous and mixed-race authors, or those who may face differential biases due to the ambiguous racialization or ethnicization of their names. We look forward to future work that could help us to better understand how to support equitable practices in science.

## Supporting information

**S1 Fig. Additional information of the intersectionality of eLife's editorial team, retrospective analysis. A.** Women Reviewing Editors ($N = 397$) serve on average slightly fewer months per year as active BRE members than men ($N = 826$) do, throughout 2017–2019. **B.** Senior Editor gender base rate. In 2021 there were significantly more men ($N = 53$) than women ($N = 30$) Senior Editors, as indicated by the asterisk. **C.** Men and women Reviewing Editors career stage. Compared to men REs, women REs were at earlier career stages, as indicated by asterisks. Note that these findings are based on data that was sampled at a different time point than our main datasets, and thus cannot be directly linked to the main findings. **D.** Reviewing Editor continent of residence. Numbers indicate the mean number of women and men REs from each continent across the three datasets (February 2019, January 2020 and December 2020); dashed yellow line depicts gender balance (50%). There was no evidence for gender disparity in the geographical representation of women and men REs. **A-C.** Men-blue, women-red; $^*p \leq 0.05$, $^{**}p \leq 0.01$, $^{***}p \leq 0.001$.
(TIF)

**S2 Fig.**
(JPG)

**S1 File.**
(DOCX)

## Acknowledgments

We thank the eLife editorial team for facilitating this study at all stages, and in particular, to: Andy Collings and Daniel Ecer for help with preparing the dataset; Jennifer Raymond, Chris Baker and Jon Roiser for discussions about the dataset; Stuart King for feedback on the manuscript; Michael B. Eisen, Tim Behrens and eLife Senior Leadership for feedback and ongoing support of this study.

## Author Contributions

**Conceptualization:** Tamar R. Makin.

**Data curation:** Maria J. Guerreiro.

**Formal analysis:** Tal Seidel Malkinson.

**Investigation:** Tal Seidel Malkinson, Maria J. Guerreiro.

**Methodology:** Tal Seidel Malkinson, Devin B. Terhune, Mathew Kollamkulam, Dani S. Bassett.

**Visualization:** Tal Seidel Malkinson.

**Writing – original draft:** Tal Seidel Malkinson, Tamar R. Makin.

**Writing – review & editing:** Tal Seidel Malkinson, Devin B. Terhune, Maria J. Guerreiro, Dani S. Bassett, Tamar R. Makin.

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
