## [Decision Letter · Decision Letter 0]

21 Mar 2023

PONE-D-22-28312Gender imbalances in the editorial activities of a selective journal run by academic editorsPLOS ONE

Dear Dr. Seidel Malkinson,

Thank you for submitting your manuscript to PLOS ONE. After careful consideration, we feel that it has merit but does not fully meet PLOS ONE’s publication criteria as it currently stands. Therefore, we invite you to submit a revised version of the manuscript that addresses the points raised during the review process.

Your manuscript has been reviewed by two proficient reviewers in the field. Overall, and although their assessment of the paper has been majorly positive, there are some technical and procedural flaws needing your attention. Concretely, key matters such as search criteria, a good acknowledgement and discussion of the research bias, and interpretation have been targeted by our reviewers in their reports.Please see below to find the full set of comments, queries, and suggestions provided by them.

We look forward to receiving your revised manuscript.

Kind regards,

Sergio A. Useche, Ph.D.

Academic Editor

PLOS ONE

Journal Requirements:

I have read the journal's policy and the authors of this manuscript have the following competing interests: Tamar Makin is a Senior Editor at eLife. Maria Guerreiro is part of the executive staff team of eLife. 

Reviewers' comments:

Reviewer's Responses to Questions

**Comments to the Author**

1. Is the manuscript technically sound, and do the data support the conclusions?

Reviewer #1: Yes

Reviewer #2: Partly

2. Has the statistical analysis been performed appropriately and rigorously? 

Reviewer #1: Yes

Reviewer #2: Yes

3. Have the authors made all data underlying the findings in their manuscript fully available?

Reviewer #1: Yes

Reviewer #2: Yes

4. Is the manuscript presented in an intelligible fashion and written in standard English?

Reviewer #1: Yes

Reviewer #2: Yes

5. Review Comments to the Author

Reviewer #1: The study aims to determine whether the involvement of individuals in elife’s board of review editors is subject to gender disparities at various stages of the editorial process. The study discovered gender imbalances in the various stages of the editorial process of eLife. This study is interesting, and the analysis is sound.

Major concern:

(1) It would be beneficial to state clearly why editorial month was chosen over individual editor count to calculate the gender composition of REs.

(2) Page 10 regarding the search of keywords in PubMed: How was the search done? I imagine there will be a lot of keyword overlap among REs. Welch’s t-test works for independent samples, while the search results are not necessarily independent?

(3) Regarding the gender difference in full submissions: I think it would be important to see if there is any gender difference in the percentage of initial submissions invited for full submissions handled by REs. This could be related to the observed differences in the number of full submissions handled by women and men REs.

(4) I understand that the study aims to capture the gender differences among the editorial activities of eLife. But it is hard to provide helpful insights to the related communities without knowing how the situation of eLife looks like compared with other journals, and how this relates to the outcomes of eLife’s editorial process – the publications. Does the presented gender imbalance in the editorial process make a difference in the overall scholarly communication system by favoring/disfavoring a particular group of authors?

Minor:

(1) Captions for tables vary across the manuscript

(2) Page 13 line 259 referred to Figure 2E, which was not available in the manuscript

(3) Page 32 Table 1: The bottom table seems incomplete?

Reviewer #2: Major Comment for Authors

Yours is an excellent introduction on the impact of publication in the lives of academics and consequences of bias. However, you omitted an important issue—the topic of the manuscript may influence its publication in a gender-biased way1. In addition, you have the opportunity to assess this given the open and huge database eLife has made available (to their credit).

I feel that the current tone of the manuscript is justification of the status quo. It may be that removing of paid eLife employees will decrease that perception. This is an important and serious criticism that must be rectified.

Specific necessary changes:

1. Justify inclusion of paid eLife employees in this publication’s authorship.

2. Add to this analysis a review the issue of topic as a potential source of gender-related bias in this journal’s publication record. That has only been shown in biomedical publication in general medical journals but may also be true in other aspects of science.

Reference List

1. Kalidasan D, Goshtasebi A, Chrisler J, Brown HL, Prior JC. Prospective analyses of sex/genderrelated publication decisions in general medical journals: editorial rejection of population- based women’s reproductive physiology. BMJ Open 2022. DOI: 10.1136/bmjopen-2021-057854.

6. PLOS authors have the option to publish the peer review history of their article (what does this mean?). If published, this will include your full peer review and any attached files.

Reviewer #1: No

Reviewer #2: **Yes: **Jerilynn C. Prior BA, MD, FRCPC

---

## [Author Response · Author response to Decision Letter 0]

3 Apr 2023

Response to Reviewers - See attached file

Reviewer #1: The study aims to determine whether the involvement of individuals in elife’s board of review editors is subject to gender disparities at various stages of the editorial process. The study discovered gender imbalances in the various stages of the editorial process of eLife. This study is interesting, and the analysis is sound.

Major concern:

(1) It would be beneficial to state clearly why editorial month was chosen over individual editor count to calculate the gender composition of REs.

Thanks you for pointing at this confusing issue. Since we were dealing with a dataset acquired over 3 years, we wanted to reflect fluctuations in the involvement of editors over this period. For example, some reviewing editors were enlisted close to the end of a given year, meaning they did not have the same opportunity to contribute to the editorial activities. Since during the studied period the representation of women in the BRE increased, it was particularly important to reflect partial contributions correctly. We chose months of BRE service as a practical unit that reflects partial work time:

To account for variability in BRE service contract durations, we used for this purpose the number of months in which the RE was affiliated with the BRE per year (months of BRE service; see methods section for further details).

In order to measure BRE members service contribution as a function of gender, we computed the proportion of months of service of women and men REs out of the total number of service months, thus taking into account partial work time (e.g. REs appointed in the middle of the year, being on leave).

In the revised manuscript, to avoid further confusion we changed our terminology from ‘months of BRE service’ to ‘service’, and provide full details of this measure in the methods. 

(2) Page 10 regarding the search of keywords in PubMed: How was the search done? I imagine there will be a lot of keyword overlap among REs. Welch’s t-test works for independent samples, while the search results are not necessarily independent?

In the keyword analysis, we counted the number of PubMed publications that could be associated to the proclaimed expertise of each RE. For example, the list of keywords provided by a specific RE (age-related macular degeneration, diabetic eye disease, retinopathy, ocular disease) was strung together using the OR operator (“age-related macular degeneration OR diabetic eye disease OR retinopathy OR ocular disease”) and queried against PubMed using the API, returning 215,250 search results (hits). This was a mean to estimate the scope and reach of the specific range of expertise, as provided by each of the REs. It is true that there is substantial overlap across REs, yet each individual was invited to provide their own set of keywords, meaning no two REs had the identical combination of terms. It might be worth highlighting that there was no initial database of terms, or suggested terms provided by eLife, meaning each RE was asked to come up with their own string of terms. 

Because keywords for expertise were self-defined, the specific terminology used by men and women could have been relevant for our research. Indeed, there is other research to show that men tend to use broader terms then women when discussing their expertise (Kolev et al., 2019). For example, given expertise in visual processing, the term ‘perception’ will have a broader scope than the word ‘vision’. We therefore felt obliged to determine whether this factor could have contributed to authors’ tendency to suggest more man than women as potential BREs at the Initial Submission stage.

The number that we fed into the Welch test was the unique number of PubMed publications that each RE query yielded. As such, these units were independent from one another, even if the same term (e.g. vision) was used as part of multiple strings. For this reason, we are not concerned about violation of the assumption of the statistical test. 

In the revised submission, we have taken a few steps to improve how this information is being communicated.

(1) We provided some further details of the operation in the results section:

Specifically, previous research points at potential disparity with the broadness of term women and men use when communicating research (Kolev et al., 2019). Accordingly, we quantified the number of PubMed search results for women and men BRE members’ keywords. Specifically, we queried the number of PubMed publications associated with the string of keywords provided by each RE, using an ‘OR’ operator. This provided us with a simple mean of scope. 

(2) We added missing information in the methods, when describing the dataset, about the fact that the keywords were self-generated:

In addition, the editorial staff asks REs to provide a set of keywords that reflect the scope of their research, which was also included in this dataset. Note that the terminology was self-generated by the Res (rather than adopted from an existing database), and that there were no limitations on the number of keywords each RE could provide.

(3) We separated the keyword analysis using a separate sub header in the methods to make it easier for the reader to obtain the relevant information. 

(3) Regarding the gender difference in full submissions: I think it would be important to see if there is any gender difference in the percentage of initial submissions invited for full submissions handled by REs. This could be related to the observed differences in the number of full submissions handled by women and men REs.

We agree with the reviewer that it would have been highly informative to learn if the decision to send a manuscript for review could have been affected by the gender of the REs assessing the Initial Submission. However, this is not a simple question to quantify based on our available database. First, we were working with two different datasets, one of which was based on individual manuscripts, the other on the REs. These were not linked with each other, meaning we did not have information about the REs involved in the Initial Assessment of each manuscript. In other words, we do not have the gender composition of the teams working during Initial Assessment. Beyond this technical limitation, it is important to emphasize that the initial decision is made by the SE based on a consultation with teams of REs, which are invited to contribute to the discussion. In this context, not all team members will contribute equally (this is something that we raised in the analysis pertaining to women’s availability to respond to the initial consultation requests and its trickle-down effects). For this reason, it becomes extremely complicated to assign gender contributions to group dynamics, and a further motivation for us to focus on the gender dynamics when only two editors are involved (SE and RE) during the Full Submission stage. This is a limitation of investigating a real-life process, we are simply unable to address this interesting and important question. 

(4) I understand that the study aims to capture the gender differences among the editorial activities of eLife. But it is hard to provide helpful insights to the related communities without knowing how the situation of eLife looks like compared with other journals, and how this relates to the outcomes of eLife’s editorial process – the publications. Does the presented gender imbalance in the editorial process make a difference in the overall scholarly communication system by favoring/disfavoring a particular group of authors?

Multiple studies have shown that across different scientific fields, women’s representation in editorial boards of scientific journals is smaller than men’s representation, e.g. 40% in journals in the field of Psychology and 30% in Neuroscience (Palser et al., 2022), 9% in the field of Mathematics, and 26% across the Frontiers publishing group (Helmer et al., 2017). Our findings of under-representation of women REs in eLife’s BRE (31% women REs and 36% women SEs) is in line with these findings. Hence, our unique access to gender dynamics in the internal editorial decision process of eLife’s BRE should also be pertinent to other scientific journals, who suffer from similar gender disparities among their editorial teams. 

Regarding the outcomes of the biases we identified, this is something that we are happy to speculate about, with relation to highly related publications. For example, a study involving 9,000 editors and 43,000 reviewers across the Frontiers publishing group have also shown homophily effects when assigning reviewers (Helmer et al., 2017). Moreover, a previous study of eLife editorial decisions focused on how the gender makeup of the participants in the peer-review stage – both editors and reviewers ¬¬– biases acceptance rates for men and women authors, found that all-men reviewer teams are far more likely to accept men-led manuscripts (Murray et al., 2019). Therefore, the homophilic behaviour that we observe at the stage of appointing REs, is likely to exacerbate homophily effects and increase the gender publishing gap. Crucially, as we explain the manuscript, the homophily effects work in the favour of men, due to historical and trickle-down disparities, which have been tracked all the way up to the makeup of editorial boards (Palser et al., 2022). 

In the revised manuscript we added the following paragraph to our conclusions section:

We find multiple consistent disparities across the editorial process, which culminate in the disproportional handling of submissions for peer review by men relative to women, even after taking into consideration men’s over-representation in eLife’s BRE. This effect was not a mere consequence of different gender distribution across disciplines, meaning it is not due to lack of available expert women, but rather a tendency of men SEs to favor men REs over women REs. This homophily effect is known to influence editorial decision-making, e.g. in recruitment of reviewers (Helmer et al., 2017) and in favorable evaluation of manuscript led by men (Murray et al., 2019). Therefore, it is easy to speculate that the disparity effects we observed here would be further amplified as the decision process trickles down. In other words, the gender disparity that tends to disfavor women cannot be pinpointed to a single stage in the editorial evaluation process, but should instead be viewed as a systematic accumulation of biases across multiple decision-making steps of a people-led process.

Minor:

(1) Captions for tables vary across the manuscript. 

Corrected with thanks.

(2) Page 13 line 259 referred to Figure 2E, which was not available in the manuscript

Corrected to Figure 1E, with thanks.

(3) Page 32 Table 1: The bottom table seems incomplete?

We apologize for the confusion, Table 1 bottom table is complete and shows the composition of eLife’s Manuscript dataset, which included two fields of chosen discipline terms for each manuscript, such that manuscript authors chose up to 2 descriptive discipline terms, out of a list of 18 possible terms, to describe their manuscript. This was clarified in the revised Table legend:

Manuscript Dataset: … It includes the following fields: … The two discipline terms the authors chose, out of 18 possible terms (Discipline 1 & Discipline 2; see Table 2 for details).

Reference

Kolev, J., Fuentes-Medel, Y., & Murray, F. (2019). Is Blinded Review Enough? How Gendered Outcomes Arise Even Under Anonymous Evaluation (No. 25759). National Bureau of Economic Research. https://doi.org/10.3386/w25759

Reviewer #2: 

Major Comment for Authors

Yours is an excellent introduction on the impact of publication in the lives of academics and consequences of bias. However, you omitted an important issue—the topic of the manuscript may influence its publication in a gender-biased way. In addition, you have the opportunity to assess this given the open and huge database eLife has made available (to their credit). I feel that the current tone of the manuscript is justification of the status quo. It may be that removing of paid eLife employees will decrease that perception. This is an important and serious criticism that must be rectified.

We thank the reviewer for her positive assessment. In the revised manuscript, we speculate on how our present findings link with other known homophily effects to create systematic gender biases that favour men. We paint this status quo, not to condone but to rectify. Due to the style of a research article, we are committed to adhering to a scientific style that is based on our own findings (and their relations to other published findings), and reserve our opinions to a more adequate format. Indeed, we have found that this research has evoked a range of reactions, and in fact this study was previously rejected from publication in another journal (PLoS Biology) by a reviewer who suggested that we are blaming men for women’s inadequacies:

“The authors don’t directly make this link, but just bemoan the fact that men handle more cases…If so, why were women “under-engaged” i.e. handling fewer articles? Well, the authors show (lines 164/170) that women RE’s were less likely to respond to the SEs’ initial consultation request and took longer to respond. … If so, we cannot blame the Ses for the fact that “women editors … were under-engaged in editorial activities”, and we cannot refer to this as bias.”

Our solution to this tension is constructive feedback. Throughout the manuscript we constantly provide suggestions and recommendations for how the biases identified in our study could be rectified (this is summarized in Table 3). We hope that the additional paragraph we added to the conclusions section helps clarify our position:

We find multiple consistent disparities across the editorial process, which culminate in the disproportional handling of submissions for peer review by men relative to women, even after taking into consideration men’s over-representation in eLife’s BRE. This effect was not a mere consequence of different gender distribution across disciplines, meaning it is not due to lack of available expert women, but rather a tendency of men SEs to favor men REs over women REs. This homophily effect is known to influence editorial decision-making, e.g. in recruitment of reviewers (Helmer et al., 2017) and in favorable evaluation of manuscript led by men (Murray et al., 2019). Therefore, it is easy to speculate that the disparity effects we observed here would be further amplified as the decision process trickles down. In other words, the gender disparity that tends to disfavor women cannot be pinpointed to a single stage in the editorial evaluation process, but should instead be viewed as a systematic accumulation of biases across multiple decision-making steps of a people-led process.

Specific necessary changes:

1. Justify inclusion of paid eLife employees in this publication’s authorship.

The only paid eLife employee is Maria Guerreiro, who was the Head of Journal Development at eLife. It is due to her commitment, time and hard work that we were able to conduct the study. Maria curated the data for us, a complicated process that required her to consider our unique requirements (e.g. GDPR ethics requirements). By any standard, this qualify Maria as a contributing author, and her conflict of interest was clearly stated as part of the submission process. It might reassure the Reviewer to learn that this ‘side’ project was not a part of Maria’s formal responsibilities with eLife, and that she has recently moved to a new organisation (EMBO).

The senior author of the manuscript (Professor Tamar Makin) is a Senior Editor with eLife, a role that is not a part of eLife’s executive team. Her involvement in the study did not relate to her paid responsibilities with eLife (which is handling submissions), but rather her role as a researcher with UCL and the University of Cambridge. Her in-depth understanding of eLife’s editorial process was essential for coming up with the research plan, analysis and write up of the manuscript, which she initiated. These roles qualify her as senior author, and her conflict of interest is clearly reported. 

It is important for us to highlight that while eLife has given us access to the data, they did not fund our research, nor did they provide us with any additional requirements. In fact, they have been helpful and facilitating of our aim to research gender biases in their editorial process. Nevertheless, to avoid the appearance of a conflict of interest, we decided to publish our manuscript in an outlet not related to eLife (PLoS One).

2. Add to this analysis a review the issue of topic as a potential source of gender-related bias in this journal’s publication record. That has only been shown in biomedical publication in general medical journals but may also be true in other aspects of science.

This is an excellent suggestion, however – unfortunately – due to GDPR constraints eLife were required to anonymize all data prior to sharing it with us. This means we were not privy to the topics (e.g. titles) of the manuscripts we were assessing. We did, however, receive information relating to the scientific field of each submission in our manuscript database. This allowed us to examine whether the main study finding – the homophily effect – was related to the proportion of manuscript assignment to women in each field, thus allowing us to assess this point on a coarser scale. The results of this analysis are reported in Figure 2. In short, we found a significant homophily effect of Senior Editor gender on the gender of the assigned RE in 14 out of the 18 disciplines (see Table 2 for details). These results demonstrate that gender homophily in manuscript assignment is a widespread cross-disciplinary effect, naturally spanning many different topics. We read your manuscript with interest, and have now included it in our Introduction and Discussion. 

Reference List

1. Kalidasan D, Goshtasebi A, Chrisler J, Brown HL, Prior JC. Prospective analyses of sex/gender related publication decisions in general medical journals: editorial rejection of population- based women’s reproductive physiology. BMJ Open 2022. DOI: 10.1136/bmjopen-2021-057854.

---

## [Decision Letter · Decision Letter 1]

22 May 2023

PONE-D-22-28312R1Gender imbalances in the editorial activities of a selective journal run by academic editorsPLOS ONE

Dear Dr. Seidel Malkinson,

Thank you for submitting your manuscript to PLOS ONE. After careful consideration, we feel that it has merit but does not fully meet PLOS ONE’s publication criteria as it currently stands. Therefore, we invite you to submit a revised version of the manuscript that addresses the points raised during the review process. Your paper has been peer-reviewed. Overall, the comments raised by our reviewer are positive, and suggest a further set of major revisions to reconsider their decision on the manuscript. Please carefully read the feedback provided by your Reviewer # 2, and make sure to reflect a suitable set of responses in the rebuttal letter, detailing the revisions made.

We look forward to receiving your revised manuscript.

Kind regards,

Sergio A. Useche, Ph.D.

Academic Editor

PLOS ONE

Reviewers' comments:

Reviewer's Responses to Questions

**Comments to the Author**

1. If the authors have adequately addressed your comments raised in a previous round of review and you feel that this manuscript is now acceptable for publication, you may indicate that here to bypass the “Comments to the Author” section, enter your conflict of interest statement in the “Confidential to Editor” section, and submit your "Accept" recommendation.

Reviewer #1: All comments have been addressed

Reviewer #2: (No Response)

2. Is the manuscript technically sound, and do the data support the conclusions?

Reviewer #1: Yes

Reviewer #2: Partly

3. Has the statistical analysis been performed appropriately and rigorously? 

Reviewer #1: Yes

Reviewer #2: No

4. Have the authors made all data underlying the findings in their manuscript fully available?

Reviewer #1: Yes

Reviewer #2: Yes

5. Is the manuscript presented in an intelligible fashion and written in standard English?

Reviewer #1: Yes

Reviewer #2: Yes

6. Review Comments to the Author

Reviewer #1: I am satisfied with the revisions made to the manuscript. I am satisfied with the revisions made to the manuscript.

Reviewer #2: Major Comment for Authors

I understand the roles of the authors who work with eLife much better—what you put in explanation to me needs to be in the manuscript, however. I also do now see that you are trying to improve equity. But it still feels to me like there is bias in the reporting/writing and the interpretation of these results. Likewise, the conclusions in Table 3 are too non-specific and vague to be of any real assistance in achievement of equity.

Specific Necessary Changes

1. Please add the explanations (like you made in response to my critique) of the roles of the authors that are officially affiliated with eLife.

2. In all statistical analysis, use judgement in interpreting the data based on the numbers in the analysis and an a priori understanding of what is important. For example, the paper is littered with references to “less responsive” women. That is based, in part, on this statistic: A response of 88% by women versus by men of 91% does not seem important to me (line 201). When small differences are statistically significant in a large data, that conclusion must be qualified.

3. The “unresponsive women” narrative is also based on this statistically significant difference in the time to respond to the request: 1.83±1.55 days vs. 1.54±1.23 days (line 203). Would a rational person consider three tenth of a day an important delay?

4. The observation that women editors are earlier in their careers is key and should be used in interpretation of all of the data. Also, it essential, to compare the time and contributions of the senior women academics with senior men academics. That doesn’t appear to have been done and must be.

5. You could compare variables between the early and mid-career women and same academic-stage men, but should acknowledge that results may be biased by urgent domestic responsibilities that younger women often have.

6. Another place where this paper itself feels gender biased is in the homophily responses by the different disciplines, it is true that it is present in both men and women editors. However, this tendency to choose men if you are a man editor is much greater (77%) than a women’s tendency to choose women (41%)-- lines 274-275. No comment is made that this proportion of assignment to same-sex reviewers in statistically different—but it certainly looks like it is! Please correct.

7. I like the constructive idea that the duration of a review needs to allow time for responses from both women and men.

7. PLOS authors have the option to publish the peer review history of their article (what does this mean?). If published, this will include your full peer review and any attached files.

Reviewer #1: No

Reviewer #2: **Yes: **Jerilynn C. Prior

---

## [Author Response · Author response to Decision Letter 1]

5 Jun 2023

Reviewer #2: Major Comment for Authors

I understand the roles of the authors who work with eLife much better—what you put in explanation to me needs to be in the manuscript, however. I also do now see that you are trying to improve equity. But it still feels to me like there is bias in the reporting/writing and the interpretation of these results. Likewise, the conclusions in Table 3 are too non-specific and vague to be of any real assistance in achievement of equity.

Reply

We thank the reviewer for her sincere efforts to help us improve the manuscript. However, we respectfully disagree with the Reviewer’s statement that we introduced biases in the reporting, writing and the interpretation of the results. We have made every effort to faithfully interpret our findings in a transparent and balanced way, and believe our interpretation reflects the data which we carefully analyzed based on best statistical practices. Below we elaborate on how decision process, in lieu of the reviewer’s comments. 

Regarding Table 3, we acknowledge that the recommendations may lack specificity and might not fully attain equity. Our intention was to provide valuable guidance for stakeholders seeking a proactive approach in pursuing that objective. In the revised manuscript, we have taken greater care to provide a clearer explanation of the purpose of Table 3, which is outlined as follows:

In the Conclusion:

Table 3 provides a summary of our results and aims to offer potential guidance to stakeholders for taking a proactive approach towards enhancing gender equity in editorial activities.

In Table 3 legend:

The proposed solutions aim to provide potential guidance to stakeholders, enabling them to adopt a proactive approach towards enhancing gender equity in editorial activities. 

Specific Necessary Changes

1. Please add the explanations (like you made in response to my critique) of the roles of the authors that are officially affiliated with eLife.

Reply

We have now included a more comprehensive conflict of interest statement, as follows:

Maria Guerreiro was the Head of Journal Development at eLife and was the sole paid eLife employee involved in this study. Her contribution was essential for conducting the research, including data curation in line with the study's specific requirements. The senior author, Professor Tamar Makin, serves as a Senior Editor at eLife. Her involvement in this study was independent of her paid responsibilities at eLife. Leveraging her understanding of eLife's editorial process, she played a vital role in developing the research plan, conducting analysis, and writing the manuscript

2. In all statistical analysis, use judgement in interpreting the data based on the numbers in the analysis and an a priori understanding of what is important. For example, the paper is littered with references to “less responsive” women. That is based, in part, on this statistic: A response of 88% by women versus by men of 91% does not seem important to me (line 201). When small differences are statistically significant in a large data, that conclusion must be qualified.

Reply

We acknowledge and appreciate the reviewer's feedback, and as a result, we have revised our language to ensure a more neutral tone. We replaced the wording “reduced responsiveness” with “longer response time and less frequent responses”, which describes the findings more closely, and is not an attribution to the women REs. 

Regarding the statistics, we assure the reviewer that careful consideration and expertise were applied in making these decisions. Several authors possess relevant expertise as statistics lecturers and advocates of best statistical practices. We acknowledge, both in this statement and in the text, that the observed effects may be subtle, as exemplified by the reviewer's provided example. However, it is worth noting that even these modest effects can have cascading consequences. For instance, a slight delay in response could potentially influence the subsequent tone set by another reviewing editor during a consultation. This is explained here:

Women took only slightly longer to respond relative to men (women were approximately 7 hours slower to answer emailed invitations), but considering the interactive nature of the consultation process, this delay could be meaningful. In the eLife initial consultation process, where editors’ interact in an on-line instant chatting format, this means that men are more likely to set the tone of the discussion by providing their opinion first, making it more difficult for women, on average, to influence the editorial decision (through conformity and anchoring cognitive biases for example [86–88]). It has been previously shown that it is more difficult to voice a different opinion once an opinion has been formed [89,90]. 

3. The “unresponsive women” narrative is also based on this statistically significant difference in the time to respond to the request: 1.83±1.55 days vs. 1.54±1.23 days (line 203). Would a rational person consider three tenth of a day an important delay?

Reply

The unfortunate truth is that a small but significant disparity in response time during a consultation carries tangible consequences downstream. Specifically, in the eLife initial consultation process, where editors’ interact in an on-line instant chatting format, this small delay means that men are more likely to set the tone of the discussion by providing their opinion first, making it more difficult for women, on average, to influence the editorial decision. This example is of utmost significance as it highlights a systemic bias that disproportionately affects women, who, as we have previously emphasized, often face greater commitments compared to men. This realization motivated us to propose concrete suggestions in Table 3: in order to provide women with an equitable opportunity to influence the editorial decision process, it is essential to take into account the delayed responses we have observed for women.

4. The observation that women editors are earlier in their careers is key and should be used in interpretation of all of the data. Also, it essential, to compare the time and contributions of the senior women academics with senior men academics. That doesn’t appear to have been done and must be.

Reply

We sincerely regret that we will not be able to accommodate this suggestion. As explained, the data we collected to demonstrate career stage was taken at a different time point and we are unable to make direct links between the effects we report in the main study. We emphasized the significance of this crucial consideration and this limitation in the Discussion in the Datasets section and the : 

However, the specific factors that we could study were not pre-determined based on our experimental needs. Accordingly, we were limited in our explanatory power, both in terms of other relevant factors that might be contributing to the observed effects (e.g., the level of seniority of each RE) and in terms of the statistical power (e.g., authors’ appeals are rather infrequent). To mitigate some of these gaps, we can gain some insight from more recent data relating to the heterogeneity of eLife’s BRE (see Supplemental Section, Fig. S1), although these recent analytics may not fully represent the dataset we analysed here. It is also important to consider the makeup of the BRE; these are invited roles, and as such, all the REs are established in their subfields. However, due to issues we expand upon below, it is possible that women REs are less senior than men REs, as described in the Supplemental data. … With these points in mind, our gender effects might be modulated by other contributing factors (such as seniority, identity that should be investigated in future research in greater detail. 

We now expanded on this point in the revised Supplementary results and Supplementary Figure 1C:

Note that these findings are based on data that was sampled at a different time point than our main datasets, and thus cannot be directly linked to the main findings.

5. You could compare variables between the early and mid-career women and same academic-stage men, but should acknowledge that results may be biased by urgent domestic responsibilities that younger women often have.

Reply

We have reiterated this important consideration in the discussion (see below). We would like to emphasize that domestic responsibilities are an important potential factor, but not the only one. Research shows that women researchers who are not overburdened with domestic responsibilities (e.g. who have advanced in their careers beyond childcare age, or who are childfree) still experience biased treatment. For example, senior women scientists tend to be overburdened with administrative responsibilities compared to their male colleagues (Winslow & Davis, 2016; Wright et al., 2003), and childfree women are negatively stereotyped, experience more workplace incivility than other employees (Verniers, 2020), and are disadvantaged in advancement in their scientific careers compared to men (Morgan et al., 2021; Santos & Dang Van Phu, 2019). Gender bias has very deep and complex roots, and we wanted the discussion to reflect the range of different causes that affect women, whoever and wherever they are. We are confident that the reviewer will support this ambition. 

Women’s longer response time, as well as less frequent response rate (by approximately 3%) could potentially be attributed to the fact that women have more duties and responsibilities than men REs. There are multiple reasons to suggest this, depending on women’s specific intersecting identities [43,91,92]. For example, senior women are overburdened by administrative responsibilities due to the institutional need to narrow the gender gap [25,30]. More specifically to our dataset, there is a hint that women REs are at an earlier career stage relative to men (Supplementary section), and hence may be more likely to have children at home than their men colleagues and thus face an added burden on their time [43], or be more laden with obtaining tenure. Another potential contributing factor is the higher standard of communication women are held to in order to receive equal acknowledgment, resulting in an imposed time-consuming quantity/quality trade-off for women, and reducing their productivity [47,93,94]. Irrespective of the reasons, our results signal that the journal submission and review process needs to shift away from monitoring decisions based on the decision time, which adds time pressure, and instead could potentially delay discussion and/or decisions about submissions until women have contributed. 

6. Another place where this paper itself feels gender biased is in the homophily responses by the different disciplines, it is true that it is present in both men and women editors. However, this tendency to choose men if you are a man editor is much greater (77%) than a women’s tendency to choose women (41%)-- lines 274-275. No comment is made that this proportion of assignment to same-sex reviewers in statistically different—but it certainly looks like it is! Please correct.

Reply

Thank you for bringing up this important point. However, the numbers cited by the reviewer do not account for the base rate of manuscript assignment to women and men REs. When considering the gender of SEs, we found that women SEs assigned to women REs 41.41% of the manuscripts, compared to 30.04% manuscript assignment to women when not taking into account the SE’s gender. Conversely, men SEs assigned to men REs 76.57% of the manuscripts, compared to 69.96% manuscript assignment to men when not taking into account the SE’s gender. This indicates an over-assignment effect of 11.37% for women, while men had an over-assignment effect of 6.61%. These effects have been statistically quantified in the contingency tables analysis (line 219), demonstrating a significant small-to-medium effect size. Importantly, this effect amplifies the disproportionate allocation of manuscripts to women REs over and above their under-representation in the BRE. We hope that this clarification puts the Reviewer at rest that we did not neglect to report any gender biases in our dataset. 

7. I like the constructive idea that the duration of a review needs to allow time for responses from both women and men.

Reply

We thank the Reviewer for her positive feedback. We hope that this adds value to the analysis we discussed in points 2-3 above.

---

## [Decision Letter · Decision Letter 2]

5 Jul 2023

PONE-D-22-28312R2Gender imbalances in the editorial activities of a selective journal run by academic editorsPLOS ONE

Dear Dr. Seidel Malkinson,

Thank you for submitting your manuscript to PLOS ONE. After careful consideration, we feel that it has merit but does not fully meet PLOS ONE’s publication criteria as it currently stands. Therefore, we invite you to submit a revised version of the manuscript that addresses the points raised during the review process.

Your paper has been re-reviewed. Overall, and although our referee highlights a reasonably good advance in the paper thanks to the set of responses and amendments provided by you, there are some relevant comments to consider (including those pointing out gender bias), which I am appending below. Please make sure to respond to all of them with all the possible detail, quality, and rigor. This way, and only if the R3 manuscript contains all these improvements made in a very suitable way, I will proceed to make a final decision on it without the need for a new review.

We look forward to receiving your revised manuscript.

Kind regards,

Sergio A. Useche, Ph.D.

Academic Editor

PLOS ONE

Reviewers' comments:

Reviewer's Responses to Questions

**Comments to the Author**

1. If the authors have adequately addressed your comments raised in a previous round of review and you feel that this manuscript is now acceptable for publication, you may indicate that here to bypass the “Comments to the Author” section, enter your conflict of interest statement in the “Confidential to Editor” section, and submit your "Accept" recommendation.

Reviewer #2: (No Response)

2. Is the manuscript technically sound, and do the data support the conclusions?

Reviewer #2: Partly

3. Has the statistical analysis been performed appropriately and rigorously? 

Reviewer #2: No

4. Have the authors made all data underlying the findings in their manuscript fully available?

Reviewer #2: Yes

5. Is the manuscript presented in an intelligible fashion and written in standard English?

Reviewer #2: Yes

6. Review Comments to the Author

Reviewer #2: PONE-D-22-28312R2 eLife evaluation of gender related to academic publication

Major Comment for Editor

Thank you for sending me the document with the minimal track changes by the authors. I understand that I need to be more specific in what I request that they do.

The responses to my comments from the authors were largely appropriate. The paper is also reviewed with the new Table (3).

However, still the same lingering rather pompous-feeling masculine bias wafts over the manuscript.

There are also some English editorial changes that are needed, such as italicizing eLife and breaking up very long and detailed paragraphs.

My suggestions need to actually be made in the manuscript and limitations listed in the Discussion. It is not sufficient that they are acknowledged in the response to reviewer document. For these reasons I’m making line-specific requirements, providing you agree, for changes in the manuscript and in the new Table 3.

I want to see this paper published. It is comprehensive and important scholarship. However, the gender bias subtleties must be respected and where they have crept in, must be removed.

Major Comment for Authors

Thank you for your positive responses to and acknowledgement of my suggestions. However, these need to make their way into the manuscript itself for the feeling of gender bias to be removed from this important piece of work.

Specific, necessary manuscript changes

1. Abstract—lines 8 to 9 Add a period after “submissions.”

Remove what is now there.

The next sentence should read: “However, women Reviewing Editors (RE) are less engaged related to these submissions, perhaps because they enter the online editorial discussions slightly (7 hours) later, and in subsequently obtained data have systematically lower academic ranks than the men RE.”

2. Abstract Line 11—SE that are men assign to men—add “(77%)”, “and likewise for women (41%).” To show that although both men and women tend to choose RE of the same sex, that phenomenon is much more prevalent in men.

3. Abstract Line 15—remove all following the word “suggest”. Add “Evidence from this intense eLife editorial gender-related review suggests that increasing the proportion of Senior Editors who are women, delaying discussions until all RE are engaged, and making all aware of the potential gender bias in men’s homophily will increase gender equity and enhance academic publication.”

4. Lines 104-6 This prediction has no referencing and sounds like a biased pre-conception. Please add evidence for this or remove it.

5. Relating to data shown on lines 200-206—there needs to be a section on Limitations in the Discussion. Add this: “Although these RE response rate differences between women and men were statistically significant, this is likely because of the large numbers in this analysis. The actual response differences between 88% of women versus 91% of men are very small. Likewise, the longer response time, again significant due to large numbers is, in reality only a delay of 7 hours for women. This is a minimal delay and unlikely in usual review circumstances to be of practical importance.”

6. Lines 210-213 re: submissions/month handled by women (.40) and men (.44). In a section on limitations in the Discussion add this: “Although women than men RE handled fewer submissions per month, common sense says that the difference between .40 and .44 manuscripts/month is unlikely to be important. This difference only reached statistical significance because of large numbers in the analysis.”

7. Referring to lines 216-222—SE referrals to RE by gender. Add to a limitations section in the Discussion: “However, the 41% that women SE assign to women RE is far less than the 77% that men SE assign to men RE. Thus, homophily is not entirely but largely a factor of men choosing other men.

8. Section describing the eLife database (Lines 310-338) needs to be added to the Introduction. It is awkward and not appropriate as part of the discussion.

9. Add to Table 3: Lines 350-351—the suggestion here (“To distribute the influence more fairly, a potential solution is to cap the number of consultations per individual RE.”)

10. Remove Lines 382-384—this is re-iterating a bias that is so minimal as to be unimportant.

11. Lines 398-399—this suggestion needs to be added to Table 3.

12. Line 444—add this suggestion to Table 3.

13. Line 463-464—add this suggested improvement to Table 3.

14. Revisions to Table 3 (as in #s 11, 12, 13) plus, separate the women with add “slightly” longer response times and the “slightly fewer women responders” into two discrete sections.

15. Revisions to Table 3 Change the wording in the remediation section on homophily to: “Increase transparency and awareness to the risks of men’s homophily on the potential gender bias in science.”

16. Table 3—change the last line of the description under the table to: “These proposed solutions aim to provide potential guidance to stakeholders, enabling them to adopt a proactive and practical approach towards enhancing gender equity in editorial activities.”

17. Line 213—start a new paragraph with the word, “Next”

18. Line 96—“data” is a plural word—please correct

19. Wherever in the document the journal, eLife, is mentioned, it should be in italics; this is standard for publications.

20. Line 467—“manuscript” should be “manuscripts”

7. PLOS authors have the option to publish the peer review history of their article (what does this mean?). If published, this will include your full peer review and any attached files.

Reviewer #2: **Yes: **Jerilynn C. Prior

---

## [Author Response · Author response to Decision Letter 2]

21 Sep 2023

Reviewer #2: PONE-D-22-28312R2 eLife evaluation of gender related to academic publication

Major Comment for Editor

Thank you for sending me the document with the minimal track changes by the authors. I understand that I need to be more specific in what I request that they do.

The responses to my comments from the authors were largely appropriate. The paper is also reviewed with the new Table (3).

However, still the same lingering rather pompous-feeling masculine bias wafts over the manuscript.

Reply

We appreciate the reviewer's constructive feedback aimed at improving our manuscript. However, we would like to express our concern about the tone and some of the comments in the review, which may appear unfounded and subjective.

Our commitment to promoting equity, diversity, and inclusion is deeply ingrained in our research objectives, and this commitment served as the driving force behind our study. While we fully acknowledge the importance of addressing gender equity in research, it is equally critical to ensure that any modifications made to the manuscript are grounded in sound statistical principles and accurately represent our findings.

We want to emphasise that the conclusions drawn from our data are firmly based on statistical analysis and should not be confused with bias. Our intention is to provide a comprehensive and data-driven perspective on the subject, and we welcome further suggestions to enhance the manuscript in this regard.

There are also some English editorial changes that are needed, such as italicizing eLife and breaking up very long and detailed paragraphs.

Reply:

We appreciate the reviewer's suggestion. However, we believe these issues can be addressed as part of the proofs with the relevant editor. 

My suggestions need to actually be made in the manuscript and limitations listed in the Discussion. It is not sufficient that they are acknowledged in the response to reviewer document. For these reasons I’m making line-specific requirements, providing you agree, for changes in the manuscript and in the new Table 3.

I want to see this paper published. It is comprehensive and important scholarship. However, the gender bias subtleties must be respected and where they have crept in, must be removed.

Reply:

Below we provide some additional clarifications to contextualise some of our key decisions concerning the revisions. Whenever possible, we have included further revisions. 

Major Comment for Authors

Thank you for your positive responses to and acknowledgement of my suggestions. However, these need to make their way into the manuscript itself for the feeling of gender bias to be removed from this important piece of work.

Specific, necessary manuscript changes

1. Abstract—lines 8 to 9 Add a period after “submissions.”

Remove what is now there.

The next sentence should read: “However, women Reviewing Editors (RE) are less engaged related to these submissions, perhaps because they enter the online editorial discussions slightly (7 hours) later, and in subsequently obtained data have systematically lower academic ranks than the men RE.”

Reply:

We appreciate the reviewer's suggestion. However, it is important to note that the information about academic rank is not directly linked to our database, and therefore, any explanation for the engagement disparities based solely on academic rank would be speculative. We do acknowledge and elaborate on this point in our Results and Discussion sections. Still, we believe that such nuances may not be suitable for the Abstract, where brevity and clarity are essential. Furthermore, in addition to the slightly longer response time, the conclusion regarding decreased engagement among women REs is also rooted in the observation of a slightly lower response rate compared to men REs. This is a factual description of our findings and by no means it assigns blame or implies any fault on the part of women. Rather, we believe it reflects a system that does not adequately account for women's characteristics and needs, as we thoroughly discuss in the later sections of the paper.

We trust that these clarifications provide a better understanding of our approach and the considerations behind our Abstract's content.

2. Abstract Line 11—SE that are men assign to men—add “(77%)”, “and likewise for women (41%).” To show that although both men and women tend to choose RE of the same sex, that phenomenon is much more prevalent in men.

Reply:

We have now incorporated this information into the Abstract, along with the assignment rates without gender specification for context. It is important to include these reference gender base rates to avoid statistical misrepresentation and provide an accurate reflection of the data.

We find evidence of gender homophily when Senior Editors assign full submissions to Reviewing Editors (i.e., men are more likely to assign full submissions to other men (77% compared to the base assignment rate to men RE of 70%), and likewise for women (41% compared to women RE base assignment rate of 30%)). 

3. Abstract Line 15—remove all following the word “suggest”. Add “Evidence from this intense eLife editorial gender-related review suggests that increasing the proportion of Senior Editors who are women, delaying discussions until all RE are engaged, and making all aware of the potential gender bias in men’s homophily will increase gender equity and enhance academic publication.”

Reply:

We thank the Reviewer for this suggestion, which we have incorporated into the abstract, with a slightly different wording.

Together, our findings confirm that gender disparities exist along the editorial process and suggest that merely increasing the proportion of women might not be sufficient to eliminate this bias. Instead, measures accounting for women's circumstances and needs (e.g., delaying discussions until all RE are engaged) and raising editorial awareness may be needed to increase gender equity and advance academic publication.

4. Lines 104-6 This prediction has no referencing and sounds like a biased pre-conception. Please add evidence for this or remove it.

Reply:

This prediction is based on the entire literature reviewed in the Introduction, which shows a consistent disparity in the participation of women in different roles and processes of scientific publishing (e.g., Dworkin et al., 2020; Helmer et al., 2017; Murray et al., 2019; Palser et al., 2022). For example, Helmer and colleagues (2017) found that women participated less as editors, reviewers and authors in an analysis of the Frontiers series of journals. Moreover, they found substantial homophily in assignment of reviewers by editors and showed that homophily will persist even if numerical parity between genders is reached. Based on this evidence we predicted that this trend will also be found in eLife when examining gender balance in between-Editors interactions and in the engagement in the editorial process. 

We followed the reviewer’s suggestion and added those references to the text:

Based on the literature reviewed above [e.g., 53,54,67,74], we predicted that despite efforts to increase the involvement of women in the BRE, women’s editorial activities would be lower in comparison to men, even after taking into consideration their proportional disparity in the editorial system. 

References:

Dworkin, J. D., Linn, K. A., Teich, E. G., Zurn, P., Shinohara, R. T., & Bassett, D. S. (2020). The extent and drivers of gender imbalance in neuroscience reference lists. Nature Neuroscience, 23(8), 918–926.

Helmer, M., Schottdorf, M., Neef, A., & Battaglia, D. (2017). Gender bias in scholarly peer review. eLife, 6. doi: 10.7554/eLife.21718

Murray, D., Siler, K., Larivière, V., Chan, W. M., Collings, A. M., Raymond, J., & Sugimoto, C. R. (2019). Author-Reviewer Homophily in Peer Review.

Palser, E. R., Lazerwitz, M., & Fotopoulou, A. (2022). Gender and geographical disparity in editorial boards of journals in psychology and neuroscience. Nature Neuroscience, 25(3), 272–279. doi: 10.1038/s41593-022-01012-w

5. Relating to data shown on lines 200-206—there needs to be a section on Limitations in the Discussion. Add this: “Although these RE response rate differences between women and men were statistically significant, this is likely because of the large numbers in this analysis. The actual response differences between 88% of women versus 91% of men are very small. Likewise, the longer response time, again significant due to large numbers is, in reality only a delay of 7 hours for women. This is a minimal delay and unlikely in usual review circumstances to be of practical importance.”

Reply:

Regretfully, we cannot comply with this request, as it departs from statistical guidelines and is grounded in a speculative assumption. The core issue lies in the assumption that our findings are significant solely due to the use of a large sample. Our study strictly adheres to rigorous statistical analysis practices, and our conclusions have been derived through meticulous data examination. Dismissing the significance of our findings by attributing them solely to sample size is not aligned with established statistical guidelines.

Furthermore, our analysis encompasses the estimation of effect sizes (Hedges's g) and a comprehensive exploration of the implications of these small effects. Although these effects may be relatively small (Hedges's g = 0.20 and Hedges's g = -0.22), they hold substantial implications within the context of eLife's specific reviewing process, characterised by an online real-time discussion platform.

It is of utmost importance to recognize that our concern extends beyond this methodological aspect. Attributing the significance of our findings solely to sample size, as suggested, could potentially undermine the nuanced and valuable insights provided by our research. Such an attribution might lead to an unfounded conclusion that there is no need to improve gender balance in the editorial process, consequently jeopardising the acceptance of our proposed corrective measures. These measures, which are directly informed by these findings, have also received endorsement within the Reviewer's comments, underscoring their importance and relevance.

6. Lines 210-213 re: submissions/month handled by women (.40) and men (.44). In a section on limitations in the Discussion add this: “Although women than men RE handled fewer submissions per month, common sense says that the difference between .40 and .44 manuscripts/month is unlikely to be important. This difference only reached statistical significance because of large numbers in the analysis.”

Reply:

Please see our previous reply to comment #5.

7. Referring to lines 216-222—SE referrals to RE by gender. Add to a limitations section in the Discussion: “However, the 41% that women SE assign to women RE is far less than the 77% that men SE assign to men RE. Thus, homophily is not entirely but largely a factor of men choosing other men.

Reply:

We understand the Reviewer’s concern, but we respectfully maintain our position on this matter. As previously discussed in an earlier review round, it is crucial to contextualise the observed assignment patterns in the broader context of gender representation among Reviewing Editors (REs). Assignments made by Senior Editors (SEs) to REs should be evaluated relative to the availability of women REs in the pool. To illustrate this point, let us consider a hypothetical scenario in which there are no men REs available. In such a situation, if women SEs assign 100% of manuscripts to women REs, it would not necessarily reflect gender-related homophily but rather be a consequence of the 100% base rate of women REs. In this specific analysis, we aim to evaluate the extent of homophily beyond the effect of underrepresentation of women REs. Thus, we must consider the base assignment rate. The 41% assignment rate of women SEs to women REs should be compared to the base assignment rate of 30% to women REs, and the 77% assignment rate of men SEs to men REs should be compared to the base assignment rate of 70% to men REs. This comparison reveals an intriguing pattern: women SEs deviate more significantly from the base rate in their homophilic assignments (14%) than men SEs do (10%). This noteworthy difference is consistently reflected across different disciplines, as demonstrated in our contingency table analyses. We hope this clarification helps elucidate our perspective on this matter. We are committed to addressing any remaining concerns or questions you may have.

8. Section describing the eLife database (Lines 310-338) needs to be added to the Introduction. It is awkward and not appropriate as part of the discussion.

Reply:

While we value the Reviewer’s input, we respectfully disagree with this particular recommendation. The section in question was intentionally placed in the Discussion to address the limitations of our datasets, which is a conventional practice in scientific writing. Additionally, we would like to clarify that the full details of the eLife database, including its limitations, are already included in the Methods section. Placing this section in the Introduction may disrupt the flow and structure of the paper. By keeping the description of the eLife database in the Discussion, we ensure that readers can connect the specific details of our research to the broader context of the eLife database and understand the implications of the limitations more effectively.

9. Add to Table 3: Lines 350-351—the suggestion here (“To distribute the influence more fairly, a potential solution is to cap the number of consultations per individual RE.”)

Reply:

Done with thanks.

10. Remove Lines 382-384—this is re-iterating a bias that is so minimal as to be unimportant.

Reply:

As we discussed in response to Comment #5, we acknowledge that the effects described in these lines may appear relatively small when viewed in isolation. However, it is essential to contextualise these effects within the specific framework of eLife's reviewing process, which is characterised by an online real-time discussion platform. In this unique setting, even seemingly small biases can have substantial implications for the overall dynamics and outcomes of the review process. Given this context, we believe it is important to retain the discussion of these effects to provide a comprehensive understanding of the challenges and dynamics at play within the eLife editorial process. We hope this explanation clarifies our position on the matter.

11. Lines 398-399—this suggestion needs to be added to Table 3.

Reply:

We appreciate the Reviewer's suggestion regarding the addition of the recommendation to provide gender-specific statistics to Senior Editors about disproportional engagement by gender (Lines 398-399). We would like to clarify that this suggestion has already been incorporated into Table 3 under the entry: "provide feedback on gender imbalance patterns for Senior Editors."

In our revised table, we have expanded upon this point to enhance clarity, and it now reads as follows: provide feedback on gender imbalance patterns for Senior Editors (e.g., statistics about disproportionate RE engagement by gender).

12. Line 444—add this suggestion to Table 3.

Reply:

We thank the Reviewer for pointing this out. We added this point to the revised Table 3:

Increase the proportion of women Senior Editors

13. Line 463-464—add this suggested improvement to Table 3.

Reply:

We thank the Reviewer for pointing this out. We added this point to the revised Table 3:

diversify the network of the Senior Editors within the BRE

14. Revisions to Table 3 (as in #s 11, 12, 13) plus, separate the women with add “slightly” longer response times and the “slightly fewer women responders” into two discrete sections.

Reply:

We thank the Reviewer for spotting this omission. We corrected this point in the revised Table 3:

Women REs take slightly longer to respond to initial consultations;

Women REs respond slightly less frequently to initial consultations

15. Revisions to Table 3 Change the wording in the remediation section on homophily to: “Increase transparency and awareness to the risks of men’s homophily on the potential gender bias in science.”

Reply:

We respectfully regret that we cannot adopt this specific recommendation, as it may not fully align with the empirical findings presented in our study. As we discussed in response to Comment #7, our study reveals that homophily exists among both men and women Senior Editors, although it may stem from different drivers, as we elaborate on in the Discussion section. However, we want to emphasise unequivocally that we firmly assert that all forms of homophily, regardless of the gender of the individuals involved, represent a potential risk to the scientific community. Homophily can restrict diversity, hinder inclusivity, and diminish critical thinking—all of which are indispensable for the advancement of science. Therefore, we maintain that the existing wording appropriately conveys the overarching risk associated with homophily in science, encompassing both men's and women's homophily.

16. Table 3—change the last line of the description under the table to: “These proposed solutions aim to provide potential guidance to stakeholders, enabling them to adopt a proactive and practical approach towards enhancing gender equity in editorial activities.”

Reply:

Done with thanks.

17. Line 213—start a new paragraph with the word, “Next”

Reply:

We appreciate the suggestion to start a new paragraph with the word "Next" at Line 213. However, we would like to point out that the word "Next" already appears in the preceding line (Line 212), which serves as the introduction to the content presented in Line 213 (We next explored the effect of the Senior Editor's gender on manuscript assignment to women and men REs). Given this existing transition, we believe that beginning a new paragraph with "Next" in Line 213 may not be necessary and could potentially disrupt the flow of the text. 

18. Line 96—“data” is a plural word—please correct

Reply:

We thank the Reviewer for catching the typo at Line 96, where "data" should indeed be treated as a plural word. We would like to reassure them that we are aware of this grammatical rule, as evidenced by multiple instances throughout the text where "data" is correctly used in the plural form, such as "data were pooled" (Line 153) and "These data" (Line 181). 

19. Wherever in the document the journal, eLife, is mentioned, it should be in italics; this is standard for publications.

Reply:

Done with thanks.

20. Line 467—“manuscript” should be “manuscripts”

Reply:

Corrected with thanks (line 484).

---

## [Decision Letter · Decision Letter 3]

18 Oct 2023

PONE-D-22-28312R3Gender imbalances in the editorial activities of a selective journal run by academic editorsPLOS ONE

Dear Dr. Seidel Malkinson,

Thank you for submitting your manuscript to PLOS ONE. After careful consideration, we feel that it has merit but does not fully meet PLOS ONE’s publication criteria as it currently stands. Therefore, we invite you to submit a revised version of the manuscript that addresses the points raised during the review process.

Your paper has been re-reviewed. The remaining reviewer asks for minor revisions. I personally agree with the referee that some slight amendments are still needed. Please address them carefully and resubmit. Anyway, as the revisions are minor, after your resubmission I will proceed to make an editorial decision without asking for new rounds of review, in case I found the changes and amendments convincing and satisfactory for me.

We look forward to receiving your revised manuscript.

Kind regards,

Sergio A. Useche, Ph.D.

Academic Editor

PLOS ONE

Journal Requirements:

Reviewers' comments:

Reviewer's Responses to Questions

**Comments to the Author**

1. If the authors have adequately addressed your comments raised in a previous round of review and you feel that this manuscript is now acceptable for publication, you may indicate that here to bypass the “Comments to the Author” section, enter your conflict of interest statement in the “Confidential to Editor” section, and submit your "Accept" recommendation.

Reviewer #2: All comments have been addressed

2. Is the manuscript technically sound, and do the data support the conclusions?

Reviewer #2: Yes

3. Has the statistical analysis been performed appropriately and rigorously? 

Reviewer #2: Yes

4. Have the authors made all data underlying the findings in their manuscript fully available?

Reviewer #2: Yes

5. Is the manuscript presented in an intelligible fashion and written in standard English?

Reviewer #2: Yes

6. Review Comments to the Author

Reviewer #2: This manuscript still needs a limitation section in the discussion, as is standard in all research publications of which I am aware.

Please add:

"It is a limitation that the level of academic achievement for all of the women and men in the data were not included at the same time as the primary data analysis. This has left an implied gender equity academic status that is likely not correct. The sociology of gender-related behaviour predicts that academic stature as well as gender likely altered the outcome of meetings in which manuscripts were evaluated."

7. PLOS authors have the option to publish the peer review history of their article (what does this mean?). If published, this will include your full peer review and any attached files.

Reviewer #2: **Yes: **Jerilynn C. Prior

---

## [Author Response · Author response to Decision Letter 3]

23 Oct 2023

Rebuttal letter 

Reviewer #2: This manuscript still needs a limitation section in the discussion, as is standard in all research publications of which I am aware.

Please add:

"It is a limitation that the level of academic achievement for all of the women and men in the data were not included at the same time as the primary data analysis. This has left an implied gender equity academic status that is likely not correct. The sociology of gender-related behaviour predicts that academic stature as well as gender likely altered the outcome of meetings in which manuscripts were evaluated."

Reply: We thank the reviewer for this comment, which we have added to the revised Discussed that now reads as follows:

Unfortunately, the fact that our primary datasets lack direct information on academic attainment levels for all women and men is a limitation. This lack of information should not be interpreted to mean that academic status is equal across genders in our datasets, an assumption that is likely to be incorrect. The sociology of gendered behavior predicts that both academic status and gender likely influenced the outcome of the interaction in which manuscripts were evaluated, as discussed below.

---

## [Editor Report · Decision Letter 4]

10 Nov 2023

Gender imbalances in the editorial activities of a selective journal run by academic editors

PONE-D-22-28312R4

Dear Dr. Seidel Malkinson,

We’re pleased to inform you that your manuscript has been judged scientifically suitable for publication and will be formally accepted for publication once it meets all outstanding technical requirements.

Kind regards,

Sergio A. Useche, Ph.D.

Academic Editor

PLOS ONE

Additional Editor Comments (optional): The changes made in response to the remaining (minor) comments are sound and accurate. The paper can be accepted in this (R4) current form.

---

## [Editor Report · Acceptance letter]

28 Nov 2023

PONE-D-22-28312R4 

Gender imbalances in the editorial activities of a selective journal run by academic editors 

Dear Dr. Seidel Malkinson:

I'm pleased to inform you that your manuscript has been deemed suitable for publication in PLOS ONE. Congratulations! Your manuscript is now with our production department. 

Kind regards, 

on behalf of

Dr. Sergio A. Useche 

Academic Editor

PLOS ONE